# Enhancing Inclusive Social, Financial, and Health Services for Persons with Disabilities in Saudi Arabia: Insights from Caregivers

**DOI:** 10.3390/healthcare13151901

**Published:** 2025-08-05

**Authors:** Ghada Alturif, Wafaa Saleh, Hessa Alsanad, Augustus Ababio-Donkor

**Affiliations:** 1Department of Social Work, College of Humanities and Social Sciences, Princess Nourah bint Abdulrahman University, P.O. Box 84428, Riyadh 11671, Saudi Arabia; gaaltarif@pnu.edu.sa (G.A.); hassanad@pnu.edu.sa (H.A.); 2School of Computing, Engineering and the Built Environment, Edinburgh Napier University, 10 Colinton Road, Edinburgh EH10 5DT, UK; 3Regional Transport Research and Education Centre, Kumasi (TRECK), Department of Civil Engineering, Kwame Nkrumah University of Science & Technology, Kumasi PMB UPO, Ghana; aadonkor@knust.edu.gh

**Keywords:** people with disabilities (PWDs), caregivers, Saudi Arabia, service accessibility, structural equation modelling, vision 2030, social inclusion, disability services

## Abstract

**Background:** Social and financial services are essential for the inclusion and well-being of people with disabilities (PWDs), who often rely on family caregivers to access these systems. In Saudi Arabia, where disability inclusion is a strategic goal under Vision 2030, understanding caregiver experiences is crucial to identifying service gaps and improving accessibility. **Objectives:** This study aimed to explore caregivers’ perspectives on awareness, perceived barriers, and accessibility of social and financial services for PWDs in Saudi Arabia. The analysis is grounded in Andersen’s Behavioural Model of Health Service Use and the WHO’s International Classification of Functioning, Disability and Health (ICF) framework. **Methods:** A cross-sectional survey was conducted with 3353 caregivers of PWDs attending specialised day schools. The survey collected data on demographic characteristics, service awareness, utilisation, and perceived obstacles. Exploratory Factor Analysis (EFA) identified latent constructs, and Structural Equation Modelling (SEM) was used to test relationships between awareness, barriers, and accessibility. **Results:** Findings reveal that over 70% of caregivers lacked awareness of available services, and only about 3% had accessed them. Key challenges included technological barriers, complex procedures, and non-functional or unclear service provider platforms. Both User Barriers and Service Barriers were negatively associated with Awareness and Accessibility. Awareness, in turn, significantly predicted perceived Accessibility. Caregiver demographics, such as age, education, gender, and geographic location, also influenced awareness and service use. **Conclusions:** There is a pressing need for targeted awareness campaigns, accessible digital service platforms, and simplified service processes tailored to diverse caregiver profiles. Inclusive communication, decentralised outreach, and policy reforms are necessary to enhance service access and promote the societal inclusion of PWDs in alignment with Saudi Arabia’s Vision 2030.

## 1. Introduction, Background and the Literature Review

The United Nations Convention on the Rights of Persons with Disabilities (UNCRPD) defines individuals with disabilities as those experiencing long-term physical, mental, intellectual, or sensory impairments that, in interaction with various barriers, hinder their full and equal participation in society [1]. Approximately 15% of the global population—around 1 billion people—live with some form of disability [2]. People with disabilities (PWDs) often require support in navigating daily life and accessing essential services, with family members frequently assuming the role of primary caregivers [3,4].

Despite the growing recognition of the role caregivers play in supporting PWDs, their perspectives, particularly in relation to service accessibility, remain underexplored in the literature. Understanding these experiences is critical for developing inclusive policies and support systems that address the needs of both PWDs and their caregivers. This study addresses this gap by exploring caregivers’ perceptions of financial and social service accessibility in Saudi Arabia.

To conceptually organise the key dimensions of service access and barriers, this study draws on two theoretical frameworks: the Andersen Behavioural Model of Health Services Use [5] and the World Health Organization’s International Classification of Functioning, Disability and Health (ICF) [6].

The Andersen model suggests that the use of health and related services is influenced by three core components: predisposing characteristics, enabling resources, and perceived or evaluated need [5]. Informed by this model, the constructs Services Awareness (Awareness) and Service Accessibility (ServAcc) are conceptualised as representing enabling factors that facilitate or hinder service utilisation. Awareness captures caregiver awareness of support services such as financial aid or vocational training programs, while ServAcc reflects the structural and administrative ease or difficulty with which these services are accessed, including perceptions of availability and quality.

The ICF framework complements this by emphasising the interaction between individual health conditions and contextual factors, including environmental and personal influences, that shape functioning and participation [6]. Guided by this perspective, the constructs Service Barriers (ServBarrier) and User Barriers (UserBarrier) reflect external environmental constraints (e.g., bureaucratic processes, physical inaccessibility) and personal-level limitations (e.g., low income, lack of information), respectively.

Although these four constructs were initially derived through exploratory factor analysis, their categorisation is theoretically supported. Awareness and ServAcc, while both linked to enabling factors, are conceptually distinct: awareness is a prerequisite for access, and perceived service quality can influence uptake. Similarly, ServBarrier and UserBarrier align with the ICF’s distinction between external and personal determinants of access. This theoretical grounding enhances the construct validity of the measurement model, as recommended in the scale development literature [7,8].

Previous studies also support this conceptual structure. For example, Turnbull and Turnbull [9], Dunst et al. [10], and Scherer et al. [11] emphasise the importance of empowering families of PWDs through awareness and family-centred service models. Hashemi et al. [12] and Mitra et al. [13] demonstrate structural and attitudinal barriers in health and transport service systems, particularly in low- and middle-income countries. Other research has identified barriers to financial inclusion, such as limited financial literacy, inaccessible environments, and discriminatory policies [14,15]. Other studies emphasised the cultural perceptions of disability and caregiving, which are critical to understanding the broader context of service access and utilisation. For example, persistent misconceptions that disabilities, particularly cognitive or intellectual disabilities, are caused by familial shame may discourage families from seeking formal diagnoses or support services. Caregivers, especially mothers, could also face social judgment as caregiving is often seen as a private family responsibility rather than a shared societal obligation. Such stigma can lead to delays in diagnosis, underutilisation of services, and reluctance to engage with public support systems. These situations highlight the importance of culturally sensitive policy and outreach strategies that not only expand service provision but also challenge deep-rooted social attitudes toward disability and caregiving [4].

In Saudi Arabia, similar issues have been identified. Alqahtani and Al-Jifree [16] examine barriers to social inclusion related to education and employment, while Alshaigi and Naji [17] highlight limited financial access for PWDs. Alshammari et al. [18] further explore employment discrimination and the lack of accommodations in the workplace and further explored employment discrimination and the lack of accommodations in the workplace. Other research [19,20] emphasise healthcare access challenges and stigma-related service limitations.

Despite ongoing reforms and increasing attention to disability inclusion in Saudi Arabia under Vision 2030, caregivers of people with disabilities (PWDs) continue to face considerable challenges in navigating social and financial support systems. While several programs and services exist, access to and effective utilisation of these services remain uneven and poorly understood from the perspective of caregivers, who often act as critical intermediaries in securing support for PWDs. This study seeks to address a central research question: What are the perspectives and experiences of caregivers of PWDs in Saudi Arabia regarding the accessibility and effectiveness of financial and social services? To investigate this question, we adopt Andersen’s Behavioural Model of Health Service Use as a guiding conceptual framework. This model emphasises the role of predisposing, enabling, and need-related factors in shaping service utilisation behaviours. In doing so, we aim to explore how individual caregiver characteristics (e.g., age, gender, education), environmental barriers, and systemic factors influence both awareness of and access to disability-related services. Using a large-scale survey of 3353 caregivers of PWDs attending specialised day schools across the Kingdom, we apply a mixed-methods quantitative approach. We first use Exploratory Factor Analysis (EFA) to identify latent constructs related to service awareness, accessibility, and user- or system-level barriers. These constructs are then tested using Structural Equation Modelling (SEM) to analyse the relationships among key factors, providing a robust empirical basis to understand accessibility challenges and their policy implications.

### Study Context: Saudi Arabia and Vision 2030

The study was conducted within the Kingdom of Saudi Arabia (KSA), a Middle Eastern country known for its vast oil reserves and significant geopolitical influence in the region, with Riyadh as its capital. The population is predominantly Arab and Sunni Muslim, with a significant expatriate community. Saudi society is deeply rooted in Islamic traditions, though recent reforms under the country’s Vision 2030 aim to diversify the economy and promote sectors like tourism and technology, alongside enhancing women’s rights and participation [21].

Vision 2030 also aims to provide equal opportunities for everyone, including PWDs, who make up 7.1% of the Saudi population [22]. The vision seeks to empower PWDs by ensuring their rights to health and personal care, rehabilitation services, and education through inclusive education, specialised services, and accessible infrastructure to ensure their independence and effective participation in Saudi society. This reflects the Kingdom’s commitment to improving the quality of life for all its citizens to receive financial, social, and job opportunities [21].

According to statistics, in Saudi Arabia, there are 355,289 individuals with hearing disabilities, 30,155 with hyperactivity and distractions, 136,833 with motor disabilities, 23,282 with autism spectrum disorder, 19,428 with Down syndrome, and 610 with visual impairments, totalling 2,036,966 individuals with disabilities in the country [22]. This highlights the importance of implementing programs that protect and support their health, social care, education, and employment opportunities. In line with this, the 2030 Vision for Support to Persons with Disabilities has initiated several projects to promote their rights and enhance services for them.

Saudi Arabia’s Vision 2030 is a transformative initiative aimed at diversifying the economy and enhancing the quality of life for all citizens, including PWDs. The vision emphasises inclusivity, accessibility, and empowerment to ensure that PWDs can participate fully in society across multiple dimensions. Additionally, Vision 2030 aims to enhance the quality and reach of social services by increasing financial assistance, expanding disability pensions and housing aid, improving rehabilitation services and specialised medical care, and addressing the stigma around disabilities through public education about PWD rights. Policy and governance reforms have also been considered to further strengthen disability rights through legislation and enforcement.

By integrating disability inclusion into its national agenda, Vision 2030 aims to create a more equitable and empowered society for PWDs. The success of these initiatives will depend on policy implementation, private sector collaboration, and sustained public awareness efforts.

This study distinguishes itself by focusing on the perspectives of carers and caregivers of PWDs in Saudi Arabia, a viewpoint that has been underrepresented in existing research. By employing a mixed-methods approach, the study integrates qualitative insights from caregivers with quantitative data analysis, offering a comprehensive understanding of the challenges faced in accessing social and financial services. The methodology provides a holistic view of the systemic barriers impacting both PWDs and their support networks.

Furthermore, the study aligns with Saudi Arabia’s Vision 2030, which emphasises social inclusion and support for PWDs. The research evaluates the effectiveness of initiatives such as the national strategy featuring 23 initiatives for PWDs, the national program for the diagnosis and classification of disabilities, and the creation of a unified national registry and statistics database. By assessing these initiatives from the caregivers’ perspectives, the study offers valuable insights into their implementation and impact, contributing to the ongoing efforts to enhance service accessibility and inclusivity for PWDs in the Kingdom.

## 2. Material and Methods

### 2.1. Introduction

This study draws upon two foundational models to guide its conceptual and analytical structure: the Andersen Behavioral Model of Health Services Use [5] and the World Health Organization’s International Classification of Functioning, Disability and Health (ICF) [6]. These frameworks informed the identification and interpretation of the four constructs derived through exploratory factor analysis: Awareness of Services (Awareness), Service Barriers (ServBarrier), User Barriers (UserBarrier), and Service Accessibility (ServAcc).

Consistent with Andersen’s model, Awareness and ServAcc represent enabling resources and perceived dimensions of access, which facilitate or hinder service utilisation. ServBarrier and UserBarrier correspond to external and personal contextual factors, respectively, which can constrain access, reflecting structural and individual-level challenges. The ICF framework further supports this categorization by highlighting how both environmental and personal elements interact to shape disability-related experiences and influence service engagement. Together, these theoretical foundations enhance the construct validity of our measurement model and ensure alignment with internationally recognized approaches to disability and service access research.

### 2.2. Data Collection

Data collection involved structured surveys designed to gather qualitative and quantitative data on various aspects related to caregivers and persons with disabilities (PWDs). To ensure that only one caregiver per child participated, we implemented unique identifiers and screening questions at the beginning of the survey. Additionally, recruitment was managed through service organisations that helped verify eligibility and avoid duplicate participation. We have now explicitly stated these measures in the methods section for clarity. Caregivers play a pivotal role in navigating the support systems available for PWDs, making their insights invaluable for assessing service gaps and barriers. The questionnaire is implemented in Arabic, the native language of respondents in the Kingdom of Saudi Arabia (KSA), and is organised into distinct sections to collect comprehensive information aligned with the research objectives:Socio-Economic Variables: This section collects demographic and socio-economic data about the person with a disability (PWD), including age, gender, type of disability, and other relevant characteristics.Caretaker Information: This section gathers details about the primary caretaker, such as age, gender, relationship to the PWD, income level, educational background, location and type of residence, and household structure.Awareness of Available Services: This section assesses the respondent’s awareness of services accessible to PWDs, encompassing educational, financial, civil, and legal services.Accessibility of Services: This section evaluates the ease with which PWDs and their caretakers can access available services, considering factors such as physical accessibility, transportation, and communication.Acceptability of Services: This section examines the suitability and cultural appropriateness of the services provided, including the attitudes of service providers and the perceived quality of services.Barriers to Service Utilisation: This section identifies obstacles that hinder the effective use of available services by PWDs and their caretakers, such as financial constraints, lack of information, or discriminatory practices.

This structured approach ensures a comprehensive understanding of the factors influencing service utilisation among PWDs and their caretakers. In addition to the quantitative data collected from caregivers, this research also gathered qualitative insights from 70 service providers working in disability-related institutions. These qualitative data, while highly valuable, are not the focus of the current manuscript, which is centred on analysing caregiver perspectives through a quantitative context. The findings from the service providers are being analysed separately and will be presented in a forthcoming publication. This separation allows for a focused interpretation of each dataset. Future studies will aim to integrate both perspectives, caregivers and service providers, to offer a more holistic understanding of the barriers, needs, and service dynamics within the disability support system in Saudi Arabia.

### 2.3. Study Population and Sampling

The study targeted caregivers of PWDs attending specialised day schools across Saudi Arabia. These schools provide individualised education and therapeutic services, making them an ideal setting for capturing rich caregiver insights into service access and barriers. A stratified cluster sampling method was employed:

Phase 1 (Stratification): The country was divided into five regions, north, south, east, west, and central.

Phase 2 (Clustering): Educational and rehabilitation institutions within each region were treated as clusters.

Phase 3 (Random Sampling): Random selection of centres/schools, followed by random selection of caregiver respondents.

This multi-stage sampling approach yielded a final sample of 3353 respondents from 375 centres nationwide. The sampling ensured broad geographic coverage and high representativeness.

An electronic questionnaire was used to ensure anonymity and ease of participation. Informed consent was obtained digitally, and all ethical considerations, including confidentiality, voluntary participation, and data protection, were strictly observed in accordance with institutional research ethics.

### 2.4. Data Analysis

Data were analysed using SPSS version 26, AMOS version 23, and Microsoft Excel 365. The analytic strategy included descriptive and inferential statistics, Factor Analysis, and Structural Equation Modelling (SEM). Descriptive statistics were used to profile socio-demographics. An Exploratory Factor Analysis (EFA), Confirmatory Factor Analysis (CFA), and SEM were performed using SPSS and AMOS to validate the factor structure and test the hypothesised relationships among constructs. The model fit was assessed using Average Variance Extracted (AVE), Composite Reliability (CR), and Discriminant Validity through square root AVE comparison, in addition to Model Fit Indices such as CFI, TLI, RMSEA, and SRMR [23,24,25]. This robust analytical approach, supported by established psychological and methodological frameworks [26,27,28], ensured rigorous construct validation. Furthermore, the AMOS analysis was enhanced using the Master Validity plugin Tool to systematically test validity thresholds [29], providing insights into the multidimensional factors influencing service accessibility for PWDs in Saudi Arabia. In addition, the study is informed by broader literature on caregiver well-being, stigma, and inclusive practices [30,31,32,33,34], which provided conceptual grounding for interpreting key findings related to caregiver burden, systemic discrimination, and regional service disparities.

## 3. Analysis and Results

### 3.1. Descriptive

The study analysed data from 3353 responses of caregivers of PWDs of various Day Care Centres. The majority of respondents were males (62.2%), while females accounted for 37.8%. The distribution of the respondents’ age is shown in Table 1. The majority of respondents are aged between 36 and 50 years (54.2%). Education levels among respondents (care takers) were relatively high, with the majority holding a university degree (43.5%). Secondary education (31.4%) and primary education (7.9%) were also represented. This suggested that many respondents had a certain level of education. The majority of respondents resided in cities (78.5%), with those in the villages constituting the minority (21.5%). The monthly household income of respondents varied, with the majority (38%) earning between SAR 5000 and 10,000.

As shown in Figure 1a, the types of disabilities among persons with disabilities (PWDs) were varied, with Autism being the most common (34.5%), followed by mental disabilities (17.9%) and physical activity-related disabilities (14.6%). Figure 1b illustrates the relationship between caregivers and PWDs, which was predominantly familial: mothers accounted for 68.6% and fathers for 19.8%. Other caregivers included siblings (5.7%) and other relatives (4.5%). It is important to note here that while a majority of the reported primary caregivers were mothers of persons with disabilities (PWDs), a substantial portion of the survey responses were submitted by male household members. This reflects a common practice in the Saudi context, where male guardians often act as the official respondents in formal or online interactions, even when the day-to-day caregiving responsibilities are primarily undertaken by female family members. As such, the gender of the respondent does not always align with the actual caregiver’s identity. Figure 1c presents the age distribution of PWDs, with the highest proportion aged 6–10 years (41.4%), indicating a young population among those surveyed.

Figure 2 presents the level of awareness of the respondents of the available social services. The Figure shows a severe awareness gap, across various social and health services, which threatens progress toward Saudi Vision 2030 goals. To gain an in-depth understanding necessary to close this gap, the study investigates this observation further through cross-tabulations, factor analysis and structural equation modelling.

The study also revealed a general lack of awareness about educational services, such as the provision of educational vouchers in cooperation with private education for PWDs. A total of 74.7% of respondents (caretakers) indicated a lack of awareness of the availability of this service, with only 3.3% indicating they have used the service before. Older respondents exhibited significantly lower levels of awareness of these services, highlighting the need for targeted campaigns among this age group. Additionally, the study found that 73.1% of respondents were unaware of higher education and scholarship opportunities for PWDs at private colleges and universities. This observation was not considered a major issue since all the PWDs in this study are still below the higher education age. Meanwhile, there is a need for increased promotion and awareness of these opportunities among the families of PWDs for future eventualities.

The high level of knowledge gap regarding existing educational support programmes for PWDs presents a significant challenge to the achievement of Vision 2030 of Saudi Arabia, which states that “We will also enable those of our people with disabilities to receive the education and job opportunities that will ensure their independence and integration as effective members of society. They will be provided with all the facilities and tools required to put them on the path to commercial success” [35].

The survey also revealed a significant lack of awareness regarding key social and health services among caregivers, as illustrated in Figure 2. Specifically, 83% were unaware of measurement and diagnostic services, 56.9% were unaware of comprehensive rehabilitation centre services, 70.2% lacked awareness of vocational rehabilitation centre services, and 61.5% were unaware of programmes aimed at integrating PWDs into the labour market. This substantial knowledge gap not only limits access to essential support but also contributes to the continued marginalisation of persons with disabilities. Particularly concerning is the limited awareness of employment inclusion programmes, which may partly explain the disproportionately high unemployment rates among PWDs in Saudi Arabia [3]. 29.8% A comprehensive awareness campaign is therefore crucial to educate the public about the available social services and promote inclusion and accessibility in all aspects of life, ultimately bridging the gap towards a more equitable society. Moreover, the study found that 56% of respondents were unfamiliar with subsidy applications for assistive medical devices, with females and university graduates showing significantly higher usage. This suggests that there is a need for targeted awareness campaigns among males and those with lower levels of education, who may face additional barriers to accessing these services. The survey also revealed a lack of awareness about Kanaf funding for PWDs (85% unaware), protection from abuse services for PWDs (64.6% unaware), digital certification services for Autism (76.8% unaware), housing support services for PWDs (83.8% unaware), and the Ministry of Sports’ Fajr Program for sports rehabilitation (89.1% unaware). These findings highlight the need for increased awareness and education about these services among the general population and for targeted campaigns to reach those who may be most in need of these services.

The survey found that 80.1% of respondents were unaware of special offers and discounts for PWDs, with rural residents showing significantly lower awareness. Overall, the survey findings highlight the need for a comprehensive awareness campaign to educate the public about the various social and financial services available to PWDs, and to promote inclusion and accessibility in all aspects of life. All service centres could be equipped to educate or create awareness about all the available services, while also providing hotlines for seeking assistance. A more inclusive and equitable society for all could be worked towards by addressing these knowledge gaps and promoting awareness and understanding.

Modern technology, for instance, intended to facilitate access to these services was found to pose a significant barrier for 29.8% of respondents (Figure 3), with a notable 31.5% remaining indifferent. This suggested a lack of understanding or familiarity with modern technology among a substantial family of PWDs. In their study [36], the researchers made a similar observation, which indicated that technology designed to provide assistance for PWDs and their caretakers rather presented significant barriers owing to the complexities and poor design quality. Previous research has noted that the simplest of assistive technologies were more likely to be successfully integrated into the daily routines of PWDs and their caretakers [37,38,39]. The literature recommends several measures to enhance the adoption and effective use of assistive technologies by PWDs and their caretakers in accessing support services. These measures include short courses or workshops, online learning platforms, demonstrations of technology for families, hands-on practice with technology usage prior to implementation, support networks for post-implementation troubleshooting, and designated “super-users” at various levels for continued support and mentorship [40,41].

The education level of caretakers emerged as a critical factor in modern technology usage, with those with lower education more likely to struggle with modern technology. Similarly, Older respondents (51 and above) also faced significant challenges with modern technology usage. The struggle with modern technology usage due to age and low education is widely reported in the literature [42,43], emphasising the importance of age-friendly solutions or developing alternative solutions to digital technology deficiency.

Lack of transportation to service providers was another significant barrier, affecting 36.7% of respondents. University-educated individuals were less likely to face this challenge. It is believed that members of this group could be in employment and consequently have access to transportation. The availability of service providers in the area was a significant concern, with 46.9% of respondents citing fewer branches or distant locations as a barrier. Rural residents (villages) were more likely to face this challenge, emphasising the need for decentralised service provision and outreach programs to bridge the gap.

The results indicate that while the majority (45.8% of respondents) had no concerns with the websites of service providers, 23.5% of respondents noted that the websites of service providers are either incomplete or not working, while 30.7% were indifferent. Again, 33.5% of the respondents noted that the procedures and conditions for service accessibility provided by service providers were not clear. Older respondents exhibited significant agreement with the assertion, indicating the difficulties they go through in navigating the procedures and conditions before accessing the services. Similarly, urban dwellers significantly agreed with the assertion. 50.5% of respondents agreed that the conditions and procedures for service accessibility present several difficulties. This was a general assertion and independent of gender. However, it is particularly pronounced among the older age groups of 50 years and above. Additionally, 43.1% of respondents were of the view that the service providers have no knowledge of their needs, indicating gaps in service delivery. The place of residence was found to have a significant influence on this assertion. A higher proportion of urban than rural dwellers share this view.

Furthermore, the majority (59.3%) of respondents believe the services provided require some level of improvement in order to meet the needs of PWDs. More males than females share this opinion (*p*-value = 0.049) at a 0.05 significance level. Older respondents also call for the improvement of the services. In addition, 37.2% of respondents believe the current terms do not adequately reflect the emerging socio-economic contexts (see Figure 4), with the influence of gender on this assertion being statistically insignificant (*p*-value = 0.264).

### 3.2. Factor Analysis and Measurement Model Development

To explore the underlying dimensions of caregivers’ perceptions regarding access to services for persons with disabilities (PWDs), an exploratory factor analysis (EFA) was conducted using SPSS (version 26.0). Principal Component Analysis (PCA) with Promax rotation was employed to extract latent constructs from the attitudinal items. This analytical strategy aligns with the Andersen Behavioural Model of Health Services Use [5], which organises determinants of service utilisation into predisposing factors, enabling resources, and perceived or evaluated needs. The extracted components reflect this structure, capturing both barriers and facilitators relevant to caregivers’ service experiences.

Table 2 presents the results of the exploratory factor analysis (EFA). The EFA yielded a four-factor solution that accounted for 50.27% of the total variance, indicating a moderate but interpretable model fit [8]. The Kaiser-Meyer-Olkin (KMO) measure of sampling adequacy (value = 0.812) and Bartlett’s Test of Sphericity (χ^2^(153) = 1342.56, *p* < 0.001) confirmed the suitability of the data for factor analysis. Although the overall explained variance (50.27%) is below the ideal threshold of 60%, it remains within the acceptable range for social science research [44]. Given the conceptual coherence of the constructs and alignment with the Andersen model, the factor solution is retained for subsequent structural analysis. The four extracted factors are discussed below:

Awareness of Services (Awareness): Reflects the caregivers’ knowledge, understanding, and recognition of available social, financial, and support services provided for persons with disabilities (PWDs). These include government programs such as Kanaf funding, housing support, rehabilitation centres, and educational vouchers. Within the Andersen Behavioural Model, this factor aligns with “enabling resources”, which influence an individual’s ability to access health and social care [5]. Awareness is a prerequisite for effective access, as caregivers must first know what services exist before they can pursue or utilise them.

Service Barriers (ServBarrier): Refers to systemic and structural obstacles in service provision, such as unclear procedures, outdated regulations, or inefficient customer service, that hinder effective utilisation. These barriers reflect external environmental factors in Andersen’s model and the environmental context domain of the WHO ICF framework [5,6].

User Barriers (UserBarrier): Captures individual-level limitations that impede caregivers’ ability to seek or access services, including limited service knowledge, low digital literacy, or transportation difficulties. This aligns with predisposing and personal characteristics in Andersen’s framework and contextual and personal factors in the ICF.

Service Accessibility (ServAcc): Reflects caregivers’ perceived ease, availability, and quality of services across health, social, economic, and educational domains. This construct corresponds to evaluated need and enabling access in Andersen’s model and represents accessibility and acceptability within the WHO ICF framework.

Factor loadings for each variable and the Cronbach’s alpha values for each extracted construct are reported in Table 2. Discriminant and convergent validity were further examined using SPSS AMOS 23.0 to assess the robustness of extracted constructs (refer to 3.3). The reliability of the extracted constructs was established using Cronbach’s alpha values exceeding the 0.70 threshold [7] and Composite Reliability (CR) values above 0.60 [8]. Average Variance Extracted (AVE) of some of the constructs (e.g., Awareness, UserBarrier) was below the threshold of 0.50. However, as noted by [23], if CR exceeds 0.60, convergent validity can still be considered acceptable despite low AVE.

### 3.3. Measurement Model Reliability and Validity

A four-factor measurement model (Figure 5) was assessed for their validity by conducting a confirmatory factor analysis (CFA) to test the measurement model using SPSS AMOS 23.0. The model fitness was evaluated based on the indices and thresholds presented in Table 3 [8].

In addition to model fit indices in Table 3 above, other psychometric properties of the scales, such as composite reliability and validity, were examined. As shown in Table 2 above, in terms of composite reliability (CR), the scales exceed the recommended cutoff value of 0.60; thus, it is reasonable to conclude that the scales are reliable [24]. However, the Average Variance Extracted (AVE) values for the constructs were all less than the required threshold of 0.5 except Service Accessibility “ServAcc”, which had an AVE of 0.540. However, [23] argue that due to the strictness of the AVE, reliability can often be established on the basis of the CR alone. Furthermore, Table 2 shows that each of the item loadings is greater than 0.50. Therefore, on the basis of the recommendation of [23], the authors assumed good model fit, reliability, convergent validity, and discriminant validity for the measurement model.

### 3.4. Structural Model

Based on theoretical considerations, we hypothesised that Service accessibility (SerAcc) will be influenced by service awareness (Awareness), service barriers (ServBarrier) and user barriers (userBarrier). Hence, we model service accessibility (SerAcc) with service awareness (Awareness), service barriers (ServBarrier) and user barrier (userBarrier). The results of the analysis are also shown in Figure 6 below. The proposed model was found to fit the data satisfactorily. The fitness indices shown in Table 4 below are all within acceptable limits [23].

The squared multiple correlations were seen to have good values for the three endogenous constructs of User Barriers, Service Accessibility and Social Services in the first-order SEM.

The structural equation model demonstrated strong overall fit, as shown in Table 4, with a chi-square/df ratio of 4.338, which falls within the commonly accepted threshold of less than 5.0 for complex models [23,24,44,45,46]. Although the chi-square test can be sensitive to large sample sizes and may overestimate lack of fit, additional indices confirm the model’s robustness. Specifically, the model achieved excellent values across several criteria: CFI = 0.964, RMSEA = 0.032, GFI = 0.969, TLI = 0.961, and SRMR = 0.030, indicating good absolute and incremental fit. The model also demonstrated strong explanatory power, with squared multiple correlations (R^2^) of 0.74 for Service Accessibility, 0.56 for User Barriers, and 0.62 for Service Awareness, as shown in Table 4. These values indicate that the model explained 74%, 56%, and 62% of the variance in these constructs, respectively.

Table 5 presents the final unstandardised regression weights and associated *p*-values, confirming that all structural relationships among the latent constructs were statistically significant (*p* < 0.05). These findings support the model’s validity in capturing the factors that influence service awareness and accessibility among caregivers of persons with disabilities in Saudi Arabia.

Theoretical and Contextual Interpretation of Key Paths:

Service Barriers → Service Accessibility (β = −0.939):

As expected, systemic obstacles such as poor service coordination, bureaucratic delays, and inaccessible websites severely impair individuals’ perception of service accessibility. This aligns with the Andersen Behavioural Model, which classifies enabling resources (or lack thereof) as critical determinants of access. In the ICF framework, these reflect environmental barriers that constrain participation.

Service Barriers → User Barriers (β = +0.992):

Institutional inefficiencies directly contribute to individual-level challenges—such as difficulty reaching service centres or using digital tools—emphasising a spillover effect from system to user level. This supports the Social Ecological Model logic, which sees personal barriers as shaped by broader institutional and structural contexts.

Service Accessibility → User Barriers (β = +0.127):

Interestingly, this small but positive path suggests that even when services are more accessible, user-level barriers (e.g., limited awareness or technology discomfort) may persist. In the Saudi context, this could reflect a lag between service provision and user capability, indicating the need for user support and education to match service expansion.

User Barriers → Awareness of Services (β = −0.155):

This negative association implies that greater personal barriers correlate with reduced awareness of available services. Rather than compensatory behaviour (i.e., seeking more information when challenged), this finding likely reflects information exclusion—where individuals facing transportation, literacy, or digital access barriers are less likely to encounter or comprehend service information. This underscores a crucial access-to-information gap.

Awareness → Service Accessibility (β = +0.919):

Awareness emerged as a strong positive predictor of perceived accessibility, reinforcing the notion that information is a powerful enabler. According to Andersen’s model, awareness is a form of enabling resource, while in the ICF framework, it relates to personal factors that facilitate engagement with the environment.

## 4. Discussion

According to the model, Awareness significantly improves Service Accessibility (β = 0.919, *p* = 0.000) and is strongly undermined by user barriers (β = −0.155, *p* < 0.000), suggesting improving awareness alone is not adequate unless service barriers are addressed.

The findings reveal a widespread knowledge gap regarding available services for persons with disabilities (PWDs) across educational, health, and social domains, a trend further supported by the structural model. Specifically, the model showed that user barriers significantly predict awareness of social services (β = −0.155, *p* < 0.001). This indicates that individuals who face challenges such as unfamiliarity with modern technology or limited access to transportation are less likely to be aware of the social and financial services available to them or their dependents. A lack of awareness acts as a critical obstacle, preventing PWDs and their families from utilising potentially life-enhancing support systems.

One of the most pronounced findings was the low level of awareness regarding educational services for PWDs. For instance, 74.7% of caretakers surveyed reported being unaware of the provision of educational vouchers in cooperation with private education providers, with only 3.3% stating that they had ever accessed such services. Older respondents exhibited significantly lower levels of awareness, highlighting the need for targeted information campaigns that address the needs of different age groups, potentially through in-person channels. Similarly, 73.1% of respondents indicated they were unaware of higher education or scholarship opportunities for PWDs. While this may not seem urgent given that most PWDs in the study were below the age of eligibility for higher education, the lack of future planning is problematic and suggests a broader issue of unpreparedness among families. These gaps present a clear challenge to the realisation of Saudi Arabia’s Vision 2030, which pledges to support the educational and employment integration of individuals with disabilities to ensure their independence and full participation in society. This finding is consistent with studies such as [43,47,48], which stress the need for improved communication strategies and user support mechanisms.

The study also found alarmingly low awareness of social and health services critical to the development and inclusion of PWDs. For example, 83% of participants were unaware of diagnostic and measurement services, 56.9% were unfamiliar with comprehensive rehabilitation centre services, and 70.2% had no knowledge of vocational rehabilitation centres. Awareness of employment support programmes, such as those integrating PWDs into the labour market, was also limited, with 61.5% reporting unfamiliarity. This lack of awareness undermines access to vital support systems and perpetuates social exclusion and economic vulnerability. The poor understanding of programmes aimed at employment inclusion is particularly concerning given the disproportionately high unemployment rate among PWDs in Saudi Arabia [49].

The survey further revealed that 56% of respondents were unfamiliar with subsidy applications for assistive medical devices. Notably, usage rates were higher among females and university-educated individuals, suggesting that educational attainment plays a critical role in navigating and accessing support services. Other important services were similarly under-recognised: 85% of respondents were unaware of Kanaf funding for PWDs; 64.6% had not heard of protection from abuse services; 76.8% were unaware of digital autism certification services; 83.8% had no knowledge of housing support services; and 89.1% were unfamiliar with the Ministry of Sports’ Fajr Program for sports rehabilitation. Moreover, 80.1% of respondents were unaware of special offers and discounts available to PWDs, with rural residents reporting particularly low levels of awareness. These findings point to an urgent need for wide-reaching and inclusive information campaigns that educate the public about available services and ensure that support reaches the most vulnerable populations.

Technological barriers also emerged as a major concern. The study found that 29.8% of respondents perceived modern technology as a barrier to service access, and 31.5% were indifferent, suggesting a general lack of familiarity or confidence with digital platforms. These findings align with previous studies indicating that assistive technologies often create additional obstacles due to poor design or complexity. To overcome this, the literature recommends a range of strategies, such as short training courses, interactive demonstrations, community-based support networks, and the identification of “super-users” who can offer ongoing mentorship. Notably, older individuals and those with lower education levels faced greater difficulties using technology, reinforcing the need for age-friendly solutions and alternative non-digital service pathways.

Transportation access also remains a significant structural barrier, with 36.7% of respondents reporting difficulty reaching service providers. Those with a university education were less likely to face such issues, possibly due to greater employment and mobility resources. Additionally, 46.9% of respondents cited the limited availability or remote locations of service centres as a major hindrance. Rural residents were particularly affected, underscoring the need for decentralised service delivery and mobile outreach initiatives that can bring services directly to underserved communities.

Service-level barriers, such as unclear and outdated service procedures, are strong determinants of service accessibility, exerting a significant negative impact on access (β = −0.939, *p* < 0.001). As presented earlier, approximately 50.5% of respondents agreed that the existing conditions and procedures for accessing services present considerable challenges. This perception was generally consistent across gender groups but was notably pronounced among respondents aged 50 years and above. Furthermore, 43.1% of participants believed that service providers lack an understanding of their specific needs, highlighting critical gaps in service delivery. The detrimental impact of service-level barriers on service accessibility or social and economic inclusion of PWDs reinforces conclusions from [11,14], which argue for stronger policy interventions and advocacy efforts.

Overall, the structural model and survey findings provide strong evidence that both service-level and personal-level barriers are restricting access to critical services for PWDs and their caretakers. These challenges, particularly low awareness and accessibility, undermine the progress toward inclusion, independence, and equality envisioned in national development goals. To address this, a dual approach is required: one that improves the availability and accessibility of services while also enhancing public awareness through inclusive, targeted campaigns tailored to users’ age, education, and geographic location. Equipping service centres to offer guidance, maintain helplines, and disseminate accurate information can serve as an immediate step toward bridging the current knowledge gap and fostering a more inclusive and equitable society.

## 5. Conclusions and Policy Recommendations

### 5.1. Conclusions

This study provides compelling evidence that both systemic and personal barriers significantly hinder the ability of caregivers to access essential services for persons with disabilities (PWDs) in Saudi Arabia. Applying Andersen’s Behavioural Model, the structural model confirms that while awareness strongly predicts service accessibility (β = 0.919, *p* = 0.000), it is substantially influenced by user-level barriers (β = −0.155, *p* < 0.001) and service-level obstacles (β = −0.939, *p* < 0.001). These findings underscore the urgent need to address not only information gaps but also structural and procedural inefficiencies that restrict meaningful access to services.

The results show that digital literacy remains a major barrier, with 29.8% of respondents identifying modern technology as an obstacle and 31.5% remaining indifferent, indicating limited engagement with digital platforms. These findings are consistent with the existing literature, highlighting usability challenges, poor interface design, and lack of training as obstacles to effective adoption of assistive technologies [3,36]. Addressing this requires inclusive digital strategies such as workshops, tutorials, post-implementation support networks, and the designation of “super-users” who can mentor others [37,38,39,40,41]. The digital divide was particularly apparent among older and less-educated caregivers, as similarly documented in [41,42,43,44,45,50,51].

Transportation and geographic access also pose critical challenges. With 36.7% of participants reporting difficulty reaching service providers and 46.9% citing distant or limited service centres, particularly in rural areas, there is a clear need for decentralised service delivery and mobile outreach strategies. These findings reinforce calls from [9,15] for flexible and community-based access solutions to reduce spatial inequalities in service provision.

Service-level issues such as outdated procedures, poor customer service, and unclear eligibility criteria further exacerbate exclusion. Nearly half of the respondents (50.5%) agreed that service conditions made access difficult, while 43.1% felt that service providers did not understand their needs. These results echo previous conclusions in [11,14], which emphasise the necessity of stronger policy interventions and user-informed service frameworks. The dissatisfaction expressed, especially by older individuals and males, highlights the importance of incorporating user feedback into ongoing service reform efforts.

While 40.8% of participants described service providers as cooperative, the remaining majority were either dissatisfied or neutral, particularly among university-educated respondents. This finding underscores the disconnect between service users’ expectations and actual service performance, further validating the call for service models that are inclusive, culturally sensitive, and family-centred [6,7].

Ultimately, this study confirms that lack of awareness, compounded by personal and systemic barriers, undermines national goals of inclusion and independence for PWDs as articulated in Vision 2030. The Vision pledges to empower PWDs with access to education, employment, and community participation: “We will also enable those of our people with disabilities to receive the education and job opportunities that will ensure their independence and integration as effective members of society. They will be provided with all the facilities and tools required to put them on the path to commercial success” [34,35].

To realise this vision, national-level action is required. This includes a coordinated awareness campaign, the redesign of service portals for user-friendliness, equipping service centres to act as information hubs, and the simplification of procedures. The recommendations in this study align with those made by [12,13], which advocate for responsive, user-informed service models designed to meet the diverse and evolving needs of PWDs and their caregivers. By integrating inclusive strategies across all service levels, policymakers and providers can bridge the current accessibility gaps and move toward a truly equitable, responsive, and empowering service ecosystem for PWDs and their families in Saudi Arabia.

Moreover, this study offers a novel empirical contribution within the Gulf region by systematically examining the intersection of caregiver characteristics, service awareness, and structural barriers in shaping access to disability services in Saudi Arabia. While the global literature has examined similar dynamics, few studies in the Gulf context have employed quantitative methods such as SEM to translate subjective caregiver experiences into validated, measurable constructs. By linking these constructs to broader system-level challenges, including digital inaccessibility, procedural opacity, and limited outreach, our findings not only address a significant research gap but also inform policy development in alignment with Saudi Arabia’s Vision 2030. These insights are equally relevant for neighbouring Gulf states undergoing social service reforms, offering evidence-based direction for designing inclusive service delivery models that respond to both user-level and structural constraints [40,41,42,50]. These region-specific studies provide critical insights into the lived experiences, emotional burdens, and systemic challenges faced by caregivers of persons with disabilities in Gulf countries. For example, research from Saudi Arabia and the UAE highlights the cultural expectations surrounding family caregiving, limited access to formal support services, and the psychological strain on parents, particularly mothers, of children with disabilities. By incorporating these perspectives, our study’s discussion on service gaps, caregiver support needs, and policy recommendations becomes more contextually grounded. These findings reinforce our call for culturally responsive interventions and caregiver-inclusive policies aligned with national development goals such as Saudi Arabia’s Vision 2030. By integrating these insights, our study contributes valuable knowledge to the existing literature and provides a comprehensive understanding of the challenges faced by caregivers of PWDs in Saudi Arabia.

Finally, it should be noted that while Structural Equation Modelling (SEM) provides robust insights into the relationships between awareness, barriers, and service utilisation, the findings are correlational rather than causal. This limitation stems from the cross-sectional nature of the data, which captures a single point in time. Furthermore, no experimental intervention was implemented in this study; the term “experiment” used earlier refers solely to the statistical modelling of survey data. Future longitudinal or experimental research designs could help establish causality and further validate these relationships.

### 5.2. Policy Recommendations

Access to inclusive services is a cornerstone of social equity and national development, particularly under Saudi Arabia’s Vision 2030. This study highlights the significant barriers caregivers of PWDs face in accessing social and financial services. Findings underscore that user-related challenges (e.g., limited digital literacy), service-level barriers (e.g., complex procedures, unresponsive providers), and broader accessibility issues negatively impact awareness and service utilisation. These results align with prior studies [11,12,13,41] that highlight institutional and perceptual barriers to disability services. To support inclusive service access, the following policy recommendations are suggested:Launch culturally sensitive awareness campaigns via mosques, public schools, and healthcare centres. Establish caregiver helplines and mobile service units in underserved areas.Offer orientation workshops at hospitals and rehabilitation centres at the point of diagnosis to familiarise caregivers with available support options.Streamline the national Disability Registry and integrate it with service provider portals to reduce administrative burdens.Embed disability education into school curricula and develop community-based anti-stigma initiatives to change public perceptions and normalise service use.

These actions not only support caregivers but also contribute to the well-being of PWDs by promoting a more responsive and accessible service system. Further research should focus on improving service quality and user satisfaction, exploring comparative studies across Gulf and international contexts, and identifying best practices to inform policy development and cross-sectoral collaboration.

## 6. Limitations and Directions for Future Research

This study offers valuable insights into the experiences of caregivers of persons with disabilities (PWDs) in Saudi Arabia, particularly regarding access to social and financial services. However, several limitations should be acknowledged.

First, the sample is limited to familial caregivers of PWDs attending specialised day schools, excluding those caring for PWDs in home settings, mainstream education, rural areas, or institutional facilities. This narrow sampling frame inherently limits the external validity of the study, as it does not capture the full diversity of caregiving contexts across the country. This restricts the generalizability of findings to the broader population of caregivers in the Kingdom. Notably, migrant domestic caregivers—who constitute a large portion of daily care providers in Saudi Arabia—were not included, despite their central role in caregiving. Their exclusion presents a significant gap, as their service access challenges and awareness levels may differ substantially. Second, although the study reveals that 70.7% of caregivers were unaware of available services and only 2.3% had used them, it does not explore the underlying reasons for these low levels. Factors such as geographic disparities, digital exclusion, socio-economic status, cultural stigma, and limited outreach efforts remain underexplored. Future research should address these dimensions through qualitative interviews and targeted surveys. Specifically, we recommend:Including caregivers from home-based and rural settings, as well as migrant care workers, for broader representation.Conducting in-depth qualitative studies to understand why caregivers lack awareness or face difficulties utilising services.Investigating how caregivers receive information and assessing the effectiveness of current communication strategies.Exploring the influence of socio-economic, cultural, and regional contexts on service access and perceptions.Examining psychosocial factors, such as stigma, caregiver burden, and family dynamics, that influence service utilisation.

## Figures and Tables

**Figure 1 healthcare-13-01901-f001:**
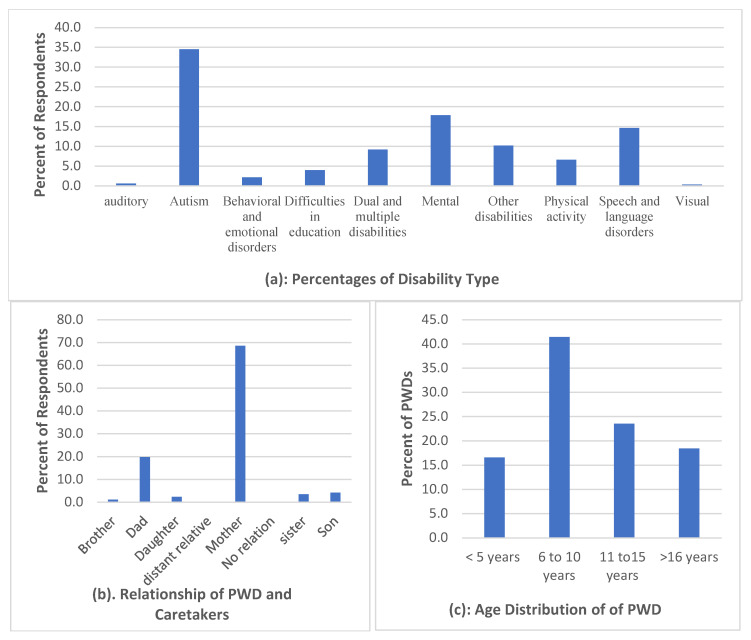
Demographics.

**Figure 2 healthcare-13-01901-f002:**
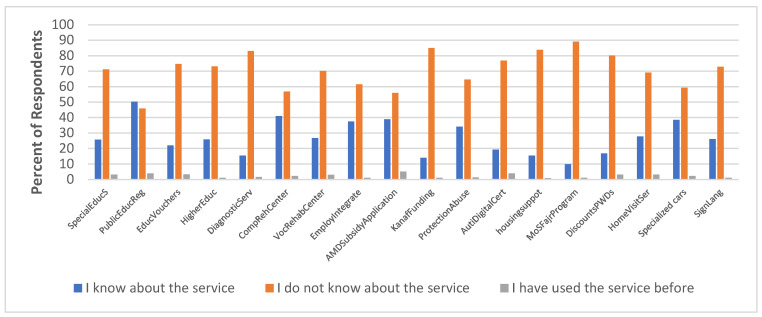
Awareness level of existing social services.

**Figure 3 healthcare-13-01901-f003:**
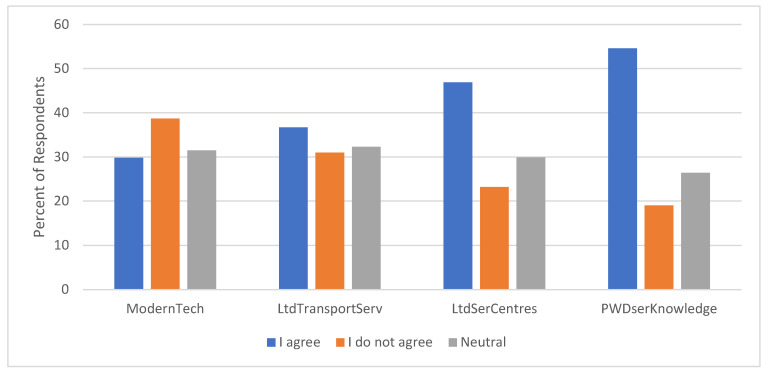
Perception of user barriers.

**Figure 4 healthcare-13-01901-f004:**
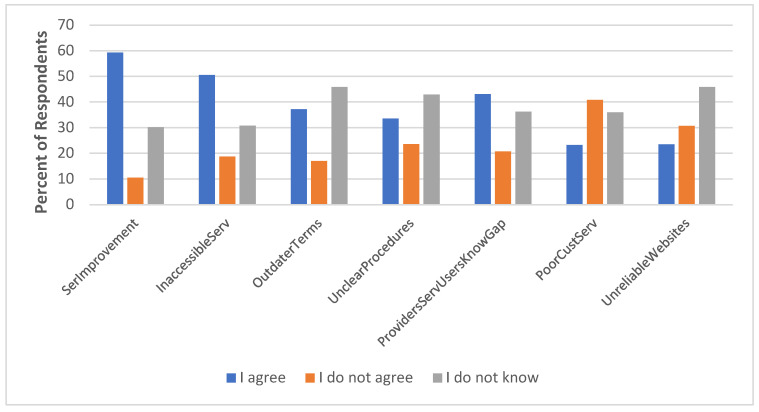
Perception of service quality.

**Figure 5 healthcare-13-01901-f005:**
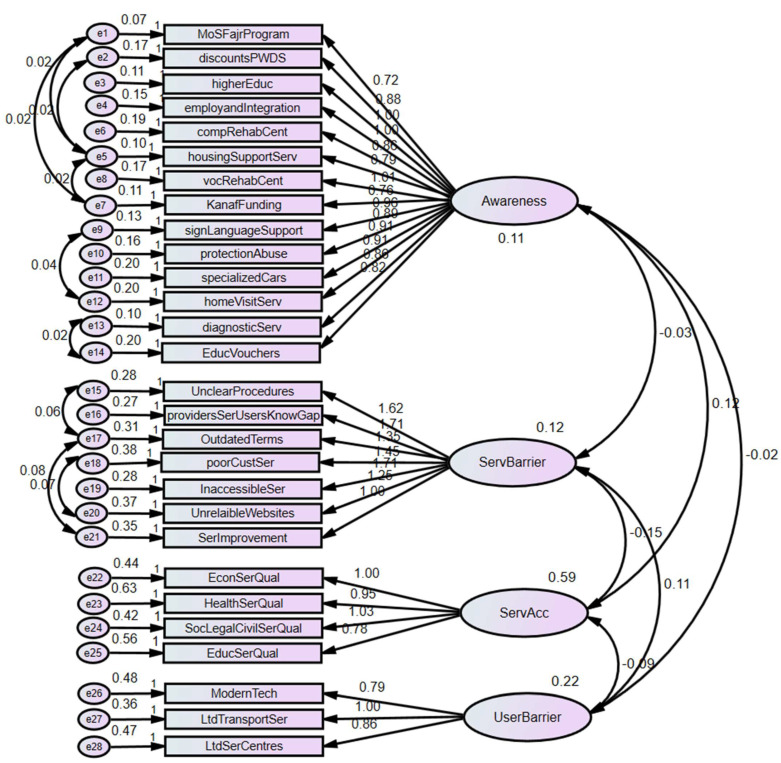
Measurement Model.

**Figure 6 healthcare-13-01901-f006:**
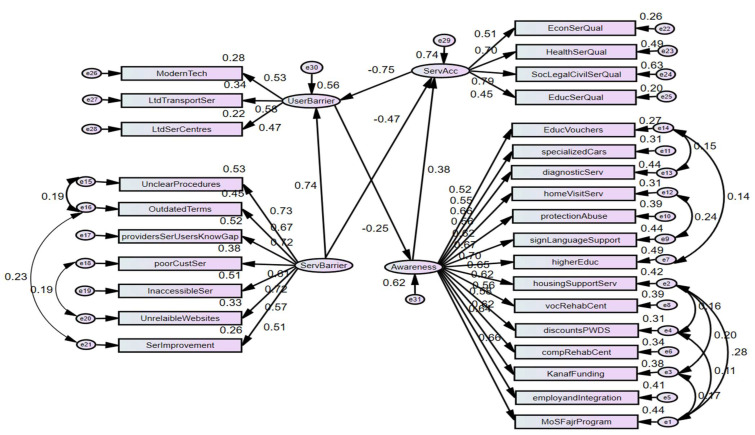
Structural Model (standardised Estimates).

**Table 1 healthcare-13-01901-t001:** Characteristics of caregivers.

Item	Description	Frequency	Percentage
Gender	Female	1268	37.8
	Male	2085	62.2
Age	Under 20	97	2.9
	21–35 yrs	992	29.6
	36–50 yrs	1816	54.2
	51 yrs and Older	448	13.4
Education	Primary	266	7.9
	Secondary	1052	31.4
	Average	347	10.3
	University	1460	43.5
	Uneducated	117	3.5
	Other	111	3.3
Income	<5000 SAR	1120	33.4
	6000–10,000 SAR	1274	38.0
	11,000–15,000 SAR	679	20.3
	>16,000 SAR	280	8.4

**Table 2 healthcare-13-01901-t002:** Factor Loadings, Reliability and Validity Indicators of the Extracted Constructs.

Construct	Code	Variable *	Item Loading	Cronbach’s Alpha (α)	Composite Reliability (CR)	Average Variance Extracted (AVE)
			>0.5	>0.7	>0.6	>0.5
Awareness (Awareness of Services)	SS1	Fajr Program	0.770	0.902	0.895	0.379
	SS2	Housing Support	0.730			
	SS3	Kanaf Funding	0.680			
	SS4	PWD Discounts	0.530			
	SS5	Employment Integration	0.620			
	SS6	Comprehensive Rehab Centre	0.540			
	SS7	Higher Education Access	0.710			
	SS8	Vocational Rehab Centres	0.620			
	SS9	Sign Language Support	0.730			
	SS10	Abuse Protection	0.562			
	SS11	Specialised Care	0.520			
	SS12	Diagnostics Services	0.720			
	SS13	Home Visit Service	0.580			
	SS14	Education Vouchers	0.520			
ServBarrier	SB1	Unclear Procedures	0.840	0.835	0.826	0.425
	SB2	Outdated Service Terms	0.791			
	SB3	Knowledge Gap in Service Use	0.723			
	SB4	Poor Customer Service	0.663			
	SB5	Physical Inaccessibility	0.647			
	SB6	Unreliable Websites	0.643			
	SB7	Lack of Service Improvement	0.630			
ServAcc	SA1	Health Services Quality	0.750	0.82	0.824	0.540
	SA2	Economic Support Services Quality	0.825			
	SA3	Social/Legal Services Quality	0.820			
	SA4	Educational Services Quality	0.778			
UserBarrier	UB1	Comfort with Modern Technology	0.710	0.624	0.638	3427
	UB2	Limited Transport Availability	0.672			
	UB3	Limited Service Centres Nearby	0.570			

* We recommend that future studies carefully identify and select constructs that not only meet statistical thresholds but are also appropriately aligned with the evolving service landscape and conceptual relevance to the local context. As service delivery systems mature and caregiver awareness improves, these constructs may yield stronger psychometric properties and offer deeper insight into access-related challenges for persons with disabilities.

**Table 3 healthcare-13-01901-t003:** Measurement Model Fit.

Model Fitness	Chi-Square Group			Absolute Fit		Incremental Fit		Standard RMR
Fitness Indexes	C_min_	*df*	C_min_/*df*	*CFI*	*RMSEA*	*GFI*	*TLI*	*SRMR*
Threshold	>0.05		<5	>0.90	<0.08	>0.90	>0.90	<0.08
This model	1504.150	335	4.490	0.964	0.032	0.968	0.959	0.027

**Table 4 healthcare-13-01901-t004:** Model Fit Indices for the Structural Equation Model.

Model Fitness	Chi-Square Group			Absolute Fit		Incremental Fit		Standard RMR
Fitness Indexes	C_min_	*df*	C_min_/*df*	CFI	RMSEA	GFI	TLI	SRMR
Threshold	>0.05		<5.0	>0.90	<0.08	>0.90	>0.90	<0.08
This model	1449.036	334	4.338	0.965	0.032	0.969	0.961	0.030

**Table 5 healthcare-13-01901-t005:** Structural Equation Model Estimates for Path Relationships Among 4, Accessibility, User Barriers, and Awareness.

			Unstandardised Estimate	*SE*	*CR*	*p*-Value
ServAcc	<---	ServBarrier	−0.939	0.046	−20.222	***
UserBarrier	<---	ServBarrier	0.992	0.051	19.599	***
UserBarrier	<---	ServAcc	0.127	0.020	6.279	***
Awareness	<---	UserBarrier	−0.155	0.016	−9.776	***
ServAcc	<---	Awareness	0.919	0.050	18.372	***

*** *p* < 0.001.

## Data Availability

The data presented in this study are available on request from the corresponding author. (The data are not publicly available due to privacy or ethical restrictions).

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
