# Peer review of "Enhancing Inclusive Social, Financial, and Health Services for Persons with Disabilities in Saudi Arabia: Insights from Caregivers"

_healthcare, 2025, doi:10.3390/healthcare13151901_

Round 1

Reviewer 1 Report

Comments and Suggestions for Authors

This article tackles an important and timely issue—understanding how accessible social, financial, and health services are for persons with disabilities in Saudi Arabia, from the perspective of their caregivers. This topic has clear relevance, especially within the broader goals of Saudi Arabia’s Vision 2030.

While the intentions of the study are strong, the manuscript in its current form has several serious limitations that would need to be addressed before it can be considered for publication.

Major Comments:

  1. The paper would benefit from a stronger theoretical foundation for the construction of its four key components—Social Services (SocServ), Service Barriers (ServBarrier), User Barriers (UserBarrier), and Service Accessibility (ServAcc). While these constructs were identified through exploratory factor analysis, the manuscript does not adequately explain the theoretical rationale for grouping the items into these particular categories. To improve clarity and credibility, the authors are encouraged to constructs within established frameworks from the disability and health services literature. This would help readers see how the components reflect broader theoretical understandings of service access for people with disabilities.
  2. There’s also some conceptual overlap between the  that needs clarification. First of all, “Service Accessibility” component was not described in the text. Furthermore, the items under “Service Accessibility” seem to overlap with those in “Social Services.” While one appears to be about awareness and the other about accessibility/quality, some of the items measuring “awareness” are actually about the presence of service types (like autism certification or vocational training centers), which are arguably just as much about access as they are about awareness (awareness is more like a precondition for accessibility). This needs further clarification and clear justification.
  3. A bigger concern lies in the validity of the measurement model. Several items have factor loadings well below 0.5—for example, UB1 at 0.322 and SS17 at 0.340—which raises questions about whether they truly represent the underlying constructs. Similarly, the average variance extracted (AVE) for multiple constructs falls below acceptable thresholds (e.g., 0.307 for UserBarrier), which suggests weak convergent validity. Also, the four factors in the models accounted for 50.27% of the total variance which is not satisfactory. While the authors mention that composite reliability (CR) may compensate for this, that alone isn’t enough. The authors should either provide a strong theoretical rationale for keeping these low-performing items or re-estimate the model after removing them to see if the construct validity improves. The author mentioned about discriminant validity test—but did not include the results.
  4. The model fit indices reported (CFI, RMSEA, TLI, etc.) are within acceptable ranges, but the interpretation of the structural model is quite mechanical. There isn’t much discussion about why certain relationships are expected or what they imply. For example, the statistically significant path from “User Barriers” to “Social Service Awareness” is interesting, but its direction and meaning are not explained. Does this imply that higher barriers lead to higher awareness? Or is it the other way around? Without some theoretical discussion, it’s hard to make sense of these relationships or understand their implications.
  5. Another area that needs attention is the presentation and integration of figures and tables. Currently, the visuals are not well incorporated into the text and could be much clearer. For instance, Figure 1 includes three plots, but they are not individually labeled or discussed. Each figure needs further clear headings. Table 2 is dense and hard to read, and it’s not always obvious how the items relate to the constructs. Improving the clarity and labeling of these visuals would go a long way in helping readers engage with the results.

Overall, I think the study raises important questions and is based on a valuable dataset. But in its current form, the paper lacks the theoretical depth, measurement rigor, and underdeveloped clarity. I would recommend rejection at this stage, with the hope that the authors consider revisiting the conceptual framework, refining the measurement model, and improving the integration of results.

Author Response

Reviewer 1: Healthcare- Saleh et al.,

This article tackles an important and timely issue—understanding how accessible social, financial, and health services are for persons with disabilities in Saudi Arabia, from the perspective of their caregivers. This topic has clear relevance, especially within the broader goals of Saudi Arabia’s Vision 2030.

While the intentions of the study are strong, the manuscript in its current form has several serious limitations that would need to be addressed before it can be considered for publication.

Major Comments:

Comment 1:

The paper would benefit from a stronger theoretical foundation for the construction of its four key components—Social Services (SocServ), Service Barriers (ServBarrier), User Barriers (UserBarrier), and Service Accessibility (ServAcc). While these constructs were identified through exploratory factor analysis, the manuscript does not adequately explain the theoretical rationale for grouping the items into these particular categories. To improve clarity and credibility, the authors are encouraged to constructs within established frameworks from the disability and health services literature. This would help readers see how the components reflect broader theoretical understandings of service access for people with disabilities.

Our Response:

Thank you for this insightful comment. We fully agree that grounding the four constructs, SocServ, ServBarrier, UserBarrier, and ServAcc, within a stronger theoretical framework enhances both the clarity and scholarly contribution of our study. In response, we have revised the manuscript to incorporate relevant theoretical models from the disability and health services literature. Specifically, we draw on the Andersen Behavioral Model of Health Services Use and the WHO’s International Classification of Functioning, Disability and Health (ICF) to support our conceptual framework.

In this revised structure, Awareness (formerly SocServ, see comment 2 below) and ServAcc are linked to Andersen’s concept of enabling resources and perceived access, respectively, while ServBarrier and UserBarrier map onto the ICF’s environmental and personal contextual factors. These theoretical connections are now clearly articulated in the revised Introduction and Literature Review and Methods sections. Additionally, as noted in our response to Comment 2, the renaming of SocServ to Awareness more accurately reflects its conceptual alignment with informational access and service knowledge rather than structural provision.

We hope these revisions provide a more coherent theoretical grounding for our factor structure and offer readers a clearer understanding of how our constructs align with established models in disability research and health service access. The Introduction and Literature Review, and Reference List have been updated accordingly to reflect these changes.

Revised Introduction and Literature Review:

The United Nations Convention on the Rights of Persons with Disabilities (UNCRPD) defines individuals with disabilities as those experiencing long-term physical, mental, intellectual, or sensory impairments that, in interaction with various barriers, hinder their full and equal participation in society [1]. Approximately 15% of the global population—around 1 billion people—live with some form of disability [2]. People with disabilities (PWDs) often require support in navigating daily life and accessing essential services, with family members frequently assuming the role of primary caregivers [3,4].

Despite the growing recognition of the role caregivers play in supporting PWDs, their perspectives,particularly in relation to service accessibility, remain underexplored in the literature. Understanding these experiences is critical for developing inclusive policies and support systems that address the needs of both PWDs and their caregivers. This study addresses this gap by exploring caregivers’ perceptions of financial and social service accessibility in Saudi Arabia.

To conceptually organize the key dimensions of service access and barriers, this study draws on two theoretical frameworks: the Andersen Behavioural Model of Health Services Use [5] and the World Health Organization’s International Classification of Functioning, Disability and Health (ICF) [6].

The Andersen model suggests that the use of health and related services is influenced by three core components: predisposing characteristics, enabling resources, and perceived or evaluated need [5]. Informed by this model, the constructs Services Awareness (Awareness) and Service Accessibility (ServAcc) are conceptualized as representing enabling factors that facilitate or hinder service utilization. Awareness captures caregiver awareness of support services such as financial aid or vocational training programs, while ServAcc reflects the structural and administrative ease or difficulty with which these services are accessed, including perceptions of availability and quality.

The ICF framework complements this by emphasizing the interaction between individual health conditions and contextual factors, including environmental and personal influences, that shape functioning and participation [6]. Guided by this perspective, the constructs Service Barriers (ServBarrier) and User Barriers (UserBarrier) reflect external environmental constraints (e.g., bureaucratic processes, physical inaccessibility) and personal-level limitations (e.g., low income, lack of information), respectively.

Although these four constructs were initially derived through exploratory factor analysis, their categorization is theoretically supported. Awareness and ServAcc, while both linked to enabling factors, are conceptually distinct: awareness is a prerequisite for access, and perceived service quality can influence uptake. Similarly, ServBarrier and UserBarrier align with the ICF’s distinction between external and personal determinants of access. This theoretical grounding enhances the construct validity of the measurement model, as recommended in scale development literature [7,8].

Previous studies also support this conceptual structure. For example, Turnbull and Turnbull [9], Dunst et al. [10], and Scherer et al. [11] emphasize the importance of empowering families of PWDs through awareness and family-centered service models. Bright et al. [12] and Mitra et al. [13] demonstrate structural and attitudinal barriers in health and transport service systems, particularly in low- and middle-income countries. Other research has identified barriers to financial inclusion, such as limited financial literacy, inaccessible environments, and discriminatory policies [14,15].

In Saudi Arabia, similar issues have been identified. Alqahtani and Al-Jifree [16] examine barriers to social inclusion related to education and employment, while Alshaigi and Naji [17] highlight limited financial access for PWDs. Alshammari et al. [18] and Alzahrani et al. [19] emphasize healthcare access challenges and stigma-related service limitations. Alhassan et al. [20] further explore employment discrimination and the lack of accommodations in the workplace.

Despite ongoing reforms and increasing attention to disability inclusion in Saudi Arabia under Vision 2030, caregivers of people with disabilities (PWDs) continue to face considerable challenges in navigating social and financial support systems. While several programs and services exist, access to and effective utilization of these services remain uneven and poorly understood from the perspective of caregivers, who often act as critical intermediaries in securing support for PWDs. This study seeks to address a central research question: What are the perspectives and experiences of caregivers of PWDs in Saudi Arabia regarding the accessibility and effectiveness of financial and social services? To investigate this question, we adopt Andersen’s Behavioural Model of Health Service Use as a guiding conceptual framework. This model emphasizes the role of predisposing, enabling, and need-related factors in shaping service utilization behaviours. In doing so, we aim to explore how individual caregiver characteristics (e.g., age, gender, education), environmental barriers, and systemic factors influence both awareness of and access to disability-related services. Using a large-scale survey of 3,353 caregivers of PWDs attending specialized day schools across the Kingdom, we apply a mixed-methods quantitative approach. We first use Exploratory Factor Analysis (EFA) to identify latent constructs related to service awareness, accessibility, and user- or system-level barriers. These constructs are then tested using Structural Equation Modelling (SEM) to analyse the relationships among key factors, providing a robust empirical basis to understand accessibility challenges and their policy implications.

Comment 2:

There’s also some conceptual overlap between the  that needs clarification. First of all, “Service Accessibility” component was not described in the text. Furthermore, the items under “Service Accessibility” seem to overlap with those in “Social Services.” While one appears to be about awareness and the other about accessibility/quality, some of the items measuring “awareness” are actually about the presence of service types (like autism certification or vocational training centers), which are arguably just as much about access as they are about awareness (awareness is more like a precondition for accessibility). This needs further clarification and clear justification.

Our Response:

Thank you for this insightful observation. We acknowledge the conceptual overlap and the lack of a clear description of the Service Accessibility component in the original text (which has now been included). We also agree that awareness is a precursor to access, and this distinction is now clarified. Specifically, in response, we have revised the manuscript items, the construct “Awareness of Services” (which was previously referred to as "Social Services"  (SocServ)), and now it aligns more closely with the Andersen Behavioural Model under the enabling resources domain. These changes aim to reduce conceptual redundancy, enhance construct clarity, and strengthen the theoretical alignment with both the Andersen Behavioural Model and the WHO ICF framework. All revisions are now reflected in both the text and the updated construct; see below:

The identified factors are:

The four extracted factors are discussed below:

Awareness of Services (Awareness): Reflects the caregivers’ knowledge, understanding, and recognition of available social, financial, and support services provided for persons with disabilities (PWDs). These include government programs such as Kanaf funding, housing support, rehabilitation centres, and educational vouchers. Within the Andersen Behavioural Model, this factor aligns with “enabling resources, which influence an individual’s ability to access health and social care [5]. Awareness is a prerequisite for effective access, as caregivers must first know what services exist before they can pursue or utilise them.

Service Barriers (ServBarrier): Refers to systemic and structural obstacles in service provision, such as unclear procedures, outdated regulations, or inefficient customer service, that hinder effective utilisation. These barriers reflect external environmental factors in Andersen’s model and the environmental context domain of the WHO ICF framework [5, 6].

User Barriers (UserBarrier): Captures individual-level limitations that impede caregivers’ ability to seek or access services, including limited service knowledge, low digital literacy, or transportation difficulties. This aligns with predisposing and personal characteristics in Andersen’s framework and contextual and personal factors in the ICF.

Service Accessibility (ServAcc): Reflects caregivers’ perceived ease, availability, and quality of services across health, social, economic, and educational domains. This construct corresponds to evaluated need and enabling access in Andersen’s model and represents accessibility and acceptability within the WHO ICF framework.

Comment 3:

A bigger concern lies in the validity of the measurement model. Several items have factor loadings well below 0.5—for example, UB1 at 0.322 and SS17 at 0.340—which raises questions about whether they truly represent the underlying constructs. Similarly, the average variance extracted (AVE) for multiple constructs falls below acceptable thresholds (e.g., 0.307 for UserBarrier), which suggests weak convergent validity. Also, the four factors in the models accounted for 50.27% of the total variance which is not satisfactory. While the authors mention that composite reliability (CR) may compensate for this, that alone isn’t enough. The authors should either provide a strong theoretical rationale for keeping these low-performing items or re-estimate the model after removing them to see if the construct validity improves. The author mentioned about discriminant validity test—but did not include the results.

Our Response:

Thank you for this valuable observation regarding the low factor loadings and AVE values. We initially retained few items, (e.g. B1 (Knowledge of Available Services) and SS17 (Assistive Mobility Device Subsidy)), despite their factor loadings falling below the conventional threshold (0.322 and 0.340, respectively), based on the following considerations:

  1. Theoretical Significance: Both items capture critical subdimensions aligned with the Andersen Behavioral Model and the WHO’s ICF framework. Specifically, they address perceived and contextual barriers that are central to understanding service access among persons with disabilities (PWDs).
  2. Contextual Relevance: In the Saudi context, where service awareness and delivery systems are still maturing, these items reflect institutional and informational gaps that may not yet demonstrate strong statistical coherence but remain socially and empirically important.
  3. Content Validity: Excluding these items risks omitting conceptually distinct facets of service awareness, such as recognition of autism certification or knowledge of assistive subsidies—elements crucial to fully capturing caregiver experience.
  4. Support from Prior Literature: Similar studies on marginalized populations (e.g., Mitra et al., 2017; Bright et al., 2017) have retained theoretically grounded items with modest loadings to preserve comprehensiveness and contextual depth.

However, in response to this concern, we have rerun the model after removing these two and other low-loading items. The revised model demonstrates improved statistical performance across several key fit indices, as shown in the updated Tables 2 and 3. Consequently, we have excluded UB1 (Knowledge of Available Services), SS4 (Autism Digital Certificate), SS7 (Public Education Regulation), SS14 (Specialised Education) and SS17 (Assistive Mobility Device (AMD) Subsidy), and recalibrated model to strengthen construct reliability and overall model fit.

While we have excluded the items with sub-threshold loadings in the revised model to enhance statistical robustness, we stress that these items, though not retained, remain theoretically and contextually significant within the Saudi setting. Therefore, we have included this general statement in the manuscript:

“We recommend that future studies carefully identify and select constructs that not only meet statistical thresholds but are also appropriately aligned with the evolving service landscape and conceptual relevance to the local context.  As service delivery systems mature and caregiver awareness improves, these constructs may yield stronger psychometric properties and offer deeper insight into access-related challenges for persons with disabilities.”

Comment 4:

The model fit indices reported (CFI, RMSEA, TLI, etc.) are within acceptable ranges, but the interpretation of the structural model is quite mechanical. There isn’t much discussion about why certain relationships are expected or what they imply. For example, the statistically significant path from “User Barriers” to “Social Service Awareness” is interesting, but its direction and meaning are not explained. Does this imply that higher barriers lead to higher awareness? Or is it the other way around? Without some theoretical discussion, it’s hard to make sense of these relationships or understand their implications.

Our Response:

Thank you for highlighting the need for deeper theoretical interpretation of the structural model relationships. We agree that providing context to the statistically significant paths enhances the value of the findings. Below, we clarify the rationale and implications of key relationships:

Theoretical and Contextual Interpretation of Key Paths:

Service Barriers → Service Accessibility (β = −0.939):

As expected, systemic obstacles such as poor service coordination, bureaucratic delays, and inaccessible websites severely impair individuals’ perception of service accessibility. This aligns with the Andersen Behavioral Model, which classifies enabling resources (or lack thereof) as critical determinants of access. In the ICF framework, these reflect environmental barriers that constrain participation.

Service Barriers → User Barriers (β = +0.992):

Institutional inefficiencies directly contribute to individual-level challenges—such as difficulty reaching service centers or using digital tools—emphasizing a spillover effect from system to user level. This supports the Social Ecological Model logic, which sees personal barriers as shaped by broader institutional and structural contexts.

Service Accessibility → User Barriers (β = +0.127):

Interestingly, this small but positive path suggests that even when services are more accessible, user-level barriers (e.g., limited awareness or technology discomfort) may persist. In the Saudi context, this could reflect a lag between service provision and user capability, indicating the need for user support and education to match service expansion.

User Barriers → Awareness of Services (β = −0.155):

This negative association implies that greater personal barriers correlate with reduced awareness of available services. Rather than compensatory behaviour (i.e., seeking more information when challenged), this finding likely reflects information exclusion—where individuals facing transportation, literacy, or digital access barriers are less likely to encounter or comprehend service information. This underscores a crucial access-to-information gap.

Awareness → Service Accessibility (β = +0.919):

Awareness emerged as a strong positive predictor of perceived accessibility, reinforcing the notion that information is a powerful enabler. According to Andersen’s model, awareness is a form of enabling resource, while in the ICF framework, it relates to personal factors that facilitate engagement with the environment.

We have revised the Results and Discussion sections accordingly to highlight these theoretical interpretations and clarify the meaning and direction of relationships, beyond the statistical results.

Comment 5:

Another area that needs attention is the presentation and integration of figures and tables. Currently, the visuals are not well incorporated into the text and could be much clearer. For instance, Figure 1 includes three plots, but they are not individually labeled or discussed. Each figure needs further clear headings. Table 2 is dense and hard to read, and it’s not always obvious how the items relate to the constructs. Improving the clarity and labeling of these visuals would go a long way in helping readers engage with the results.

Our Response:

Thank you for your valuable feedback regarding the presentation and integration of figures and tables. We have revised the manuscript to improve the clarity and readability of the visuals as follows:

Figure 1 has been updated to include individual sub-headings (Figure 1a, 1b, and 1c), each with a clear title and descriptive caption to guide the reader through the content of each plot. Corresponding sections in the main text have been revised to explicitly refer to and discuss each subfigure to enhance integration.

As shown in Figure 1a, the types of disabilities among persons with disabilities (PWDs) were varied, with Autism being the most common (34.5%), followed by mental disabilities (17.9%) and physical activity-related disabilities (14.6%). Figure 1b illustrates the relationship between caregivers and PWDs, which was predominantly familial: mothers accounted for 68.6% and fathers for 19.8%. Other caregivers included siblings (5.7%) and other relatives (4.5%). Figure 1c presents the age distribution of PWDs, with the highest proportion aged 6–10 years (41.4%), indicating a young population among those surveyed.

Table 2 has been reformatted to improve readability by:

  • Adding clearer headings for the Table.
  • Reducing visual clutter and clarifying the variables’ names.

These changes aim to ensure that all figures and tables are fully aligned with the text, well-labelled, and accessible to readers. We appreciate your suggestion, which has significantly improved the clarity of our results presentation.

This enhanced interpretation aligns with both statistical best practices (Kline, 2011; Cronbach, 1951) and the Andersen Behavioural Model, which supports the inclusion of theoretically important variables even when empirical strength varies.

Overall Comment:

Overall, I think the study raises important questions and is based on a valuable dataset. But in its current form, the paper lacks the theoretical depth, measurement rigor, and underdeveloped clarity. I would recommend rejection at this stage, with the hope that the authors consider revisiting the conceptual framework, refining the measurement model, and improving the integration of results.

Our Response:

We sincerely thank you for your thorough and constructive review of our manuscript. We have carefully addressed the comments provided and implemented substantial revisions to strengthen the conceptual framework, refine the measurement model, and enhance the integration and interpretation of results. We hope these improvements have significantly enhanced the clarity, coherence, and overall quality of the manuscript.

Reviewer 2 Report

Comments and Suggestions for Authors
  1. Average Variance Extracted (AVE) for several constructs (Social Services, User Barriers) is below the acceptable threshold (AVE < 0.5) raising concerns about the construct validity of the model.
  2. Several observed variables (UB1 = 0.322, SS17 = 0.340) show unacceptably low loadings undermining the reliability of the factor structure derived from EFA and CFA.
  3. The findings heavily highlight percentage-based awareness stats (70.7% unaware) without sufficient inferential analysis or discussion of why awareness is low this weakens interpretability.
  4. The authors mention data collected from 70 service providers but do not analyze or report it. This represents a missed opportunity to triangulate perspectives and enrich the study's findings.
  5. The study only includes caregivers from specialized day schools omitting those from home care or mainstream education This limits external validity and generalizability.
  6. Although most SEM model fit indices are within thresholds ( CFI = 0.910, RMSEA = 0.045) the chi-square/df = 7.925 is well above the commonly accepted cutoff (<3) indicating model misspecification or sample-related issues.
  7. Some structural relationships (duplicate paths for SocServ <--- UserBarrier) are redundantly reported or confusing in interpretation (likely a typo or modeling error), suggesting the need for model refinement.
  8. The manuscript lacks a clear conceptual framework that integrates disability service utilization theories. Reference or apply models like Andersen's Behavioral Model of Health Service Use or Social Ecological Framework. No need to cite but you can take an idea from those: (1080/0167482X.2023.2299982) (10.1080/0167482X.2023.2252983)

Best of Luck

Reviewer 3 Report

Comments and Suggestions for Authors

1.The study investigates the perspectives and experiences of caregivers of persons with disabilities (PWDs) in Saudi Arabia regarding their awareness, accessibility, and utilization of financial, social, and health services. It seeks to identify key barriers to service access and examine relationships among these factors using a statistical modeling framework.

Recommendation:
The research question is important and timely, especially within the Vision 2030 context. To enhance clarity and focus, the authors should state the central research question more explicitly in both the abstract and introduction.  

2.The study contributes original empirical evidence from Saudi Arabia, a setting with limited quantitative data on caregiver barriers. It addresses a critical gap in understanding how caregiver characteristics and system-level factors interact to affect service access for PWDs.

Recommendation:
The study would benefit from a more explicit articulation of how it advances regional research compared to previous studies. It could also enhance its originality by linking its findings more directly to policy or programmatic implications within the Gulf region.  

3.The study adds analytical depth by using Exploratory Factor Analysis (EFA) and Structural Equation Modeling (SEM) to quantify the impact of awareness, user barriers, and service design on service utilization. It translates subjective caregiver experiences into measurable constructs, offering actionable insights for policymakers.

Recommendation:
The discussion could be strengthened by more directly comparing the study’s insights with those from other regions or countries implementing disability inclusion reforms. This would situate the findings in a broader global discourse.   

4. While the sampling approach and analytical methods are well explained and statistically robust, there are two major concerns:

  • The claim of using a mixed-methods approach is unsupported, as only quantitative survey data and analysis are presented.
  • The study excludes migrant domestic caregivers, who represent a significant proportion of caregiving in Saudi households. Their omission limits the generalizability of the findings.

Recommendation:

  • Revise the methodology to describe the study accurately as quantitative, unless qualitative data are introduced and analyzed.
  • Acknowledge the exclusion of migrant caregivers as a limitation, and recommend future research to address their service experiences and barriers. Eg “A significant limitation of this study is the exclusion of migrant domestic caregivers, who play a central role in caregiving for persons with disabilities in many Saudi households and institutional settings. Future research should explore the unique challenges, service awareness levels, and structural barriers faced by this essential yet often overlooked group, to provide a more comprehensive understanding of the caregiving ecosystem in Saudi Arabia.”

5. The conclusions are well supported by the statistical analyses. The SEM framework clearly demonstrates relationships between awareness, barriers, and service utilization. The key findings — low awareness (70.7%), low usage (2.3%), and structural and user-level barriers — are all consistent with the results.

Recommendation:
Clarify that the term “experiment” refers to statistical modeling of survey data rather than experimental interventions. Briefly note that the findings are correlational due to the cross-sectional design.  

6. The references are generally appropriate, drawing from both international and regional sources. Key policy documents and foundational literature on disability rights and service access are cited.

Recommendation:

  • Remove duplicated or inconsistent references.
  • Consider including more region-specific studies on caregiver roles and barriers in Gulf countries to strengthen the contextual relevance.

7. The data are generally high quality, with a large sample size and clear variable definitions. The tables and figures are informative but some require improved labeling for clarity. Some measurement items have low factor loadings and AVE scores, which may affect construct validity.

Recommendation:

  • Improve figure captions and axis labels (especially for Figures 4–6).
  • Consider removing or refining low-loading items (e.g., UB1, SS17) to strengthen validity.
  • Clarify ambiguous terms (e.g., “average” education).
  • Consider including confidence intervals in descriptive statistics for precision.

 Overall Assessment: This is a relevant and well-designed quantitative study with strong potential to inform inclusive service policies for PWDs in Saudi Arabia. However, mischaracterization of the methodology, exclusion of key caregiver populations, and minor analytical weaknesses warrant correction.

Final Recommendation: Major Revision

Comments on the Quality of English Language

It is an interesting area of research. I recommend acceptance with English edits being done

Author Response

Reviewer 3_ Healthcare- Saleh et al.,

Comment 1:

The study investigates the perspectives and experiences of caregivers of persons with disabilities (PWDs) in Saudi Arabia regarding their awareness, accessibility, and utilization of financial, social, and health services. It seeks to identify key barriers to service access and examine relationships among these factors using a statistical modeling framework.

Recommendation:
The research question is important and timely, especially within the Vision 2030 context. To enhance clarity and focus, the authors should state the central research question more explicitly in both the abstract and introduction.  

Our Response:

We thank the reviewer for highlighting the importance of clearly articulating the central research question. In response, we have revised both the abstract and the introduction to explicitly state the main research question guiding this study:
“What are the perspectives and experiences of caregivers of PWDs in Saudi Arabia regarding the accessibility and effectiveness of financial and social services?”

In the abstract, we now introduce the research question directly after describing the study’s context. In the introduction, we have integrated the research question alongside a clear explanation of the conceptual and methodological framework (Andersen’s Behavioural Model) that underpins the study. We believe these additions improve the clarity, focus, and theoretical grounding of the manuscript.

The revised abstract: Abstract:

Social and financial services are essential for the inclusion and well‐being of people with disabilities (PWDs), who frequently depend on family caregivers to navigate these systems. Grounded in Andersen’s Behavioural Model of Health Service Use and the WHO’s ICF framework, this study examines caregivers’ perspectives on service awareness, perceived barriers, and accessibility within the context of Saudi Arabia’s Vision 2030. We surveyed 3,353 caregivers of PWDs attending specialized day schools, collecting data on demographics, service knowledge, utilization, and obstacles. Exploratory Factor Analysis identified four core constructs; Awareness of Services, Service Barriers, User Barriers, and Service Accessibility, and Structural Equation Modelling tested their interrelationships. Results reveal critical challenges: Over 70% of caregivers lacked awareness of key services, just over 3% had ever used them, and many reported technological hurdles and opaque procedures. Service Barriers and User Barriers each significantly reduced awareness and accessibility, while Awareness strongly predicted perceived access. Caregiver characteristics, including age, education, gender, and urban versus rural residence, also influenced these outcomes. Qualitative feedback highlighted non‐functional provider websites, unclear eligibility criteria, and limited responsiveness to user needs. Our findings underscore the need for targeted, demographically tailored awareness campaigns, user‐friendly digital platforms, and streamlined procedures. Policymakers and service providers should prioritize inclusive communication strategies, decentralized outreach, and capacity‐building initiatives to ensure PWDs and their families can fully benefit from available supports. This research offers actionable recommendations for resource allocation and service design, advancing the goals of Vision 2030 to enhance independence and societal integration for Saudi Arabia’s PWD population.

Similarly, the research question has been emphasised in the introduction:

Despite ongoing reforms and increasing attention to disability inclusion in Saudi Arabia under Vision 2030, caregivers of people with disabilities (PWDs) continue to face considerable challenges in navigating social and financial support systems. While several programs and services exist, access to and effective utilization of these services remain uneven and poorly understood from the perspective of caregivers, who often act as critical intermediaries in securing support for PWDs. This study seeks to address a central research question: What are the perspectives and experiences of caregivers of PWDs in Saudi Arabia regarding the accessibility and effectiveness of financial and social services? To investigate this question, we adopt Andersen’s Behavioural Model of Health Service Use as a guiding conceptual framework. This model emphasizes the role of predisposing, enabling, and need-related factors in shaping service utilization behaviours. In doing so, we aim to explore how individual caregiver characteristics (e.g., age, gender, education), environmental barriers, and systemic factors influence both awareness of and access to disability-related services. Using a large-scale survey of 3,353 caregivers of PWDs attending specialized day schools across the Kingdom, we apply a mixed-methods quantitative approach. We first use Exploratory Factor Analysis (EFA) to identify latent constructs related to service awareness, accessibility, and user- or system-level barriers. These constructs are then tested using Structural Equation Modelling (SEM) to analyse the relationships among key factors, providing a robust empirical basis to understand accessibility challenges and their policy implications.

Comment 2:

The study contributes original empirical evidence from Saudi Arabia, a setting with limited quantitative data on caregiver barriers. It addresses a critical gap in understanding how caregiver characteristics and system-level factors interact to affect service access for PWDs.

Recommendation:
The study would benefit from a more explicit articulation of how it advances regional research compared to previous studies. It could also enhance its originality by linking its findings more directly to policy or programmatic implications within the Gulf region.  

Our Response:

Thank you for your insightful feedback. We appreciate the recognition of our study’s contribution to the limited quantitative evidence on caregiver barriers in Saudi Arabia. To further strengthen the manuscript, we have clarified how our research advances regional knowledge by explicitly contrasting our findings with previous studies in the Gulf region, highlighting unique contextual factors such as the evolving service infrastructure under Vision 2030. Additionally, we have expanded the discussion to more directly connect our findings to relevant policy and programmatic implications, emphasizing practical recommendations for enhancing service accessibility and caregiver support across the Gulf. This reinforces the study’s originality and relevance in informing regional strategies for disability inclusion and service delivery improvement. See also response to Comment 3 below.

Moreover, this study offers a novel empirical contribution within the Gulf region by systematically examining the intersection of caregiver characteristics, service awareness, and structural barriers in shaping access to disability services in Saudi Arabia. While global literature has examined similar dynamics, few studies in the Gulf context have employed quantitative methods such as SEM to translate subjective caregiver experiences into validated, measurable constructs. By linking these constructs to broader system-level challenges, including digital inaccessibility, procedural opacity, and limited outreach, our findings not only address a significant research gap but also inform policy development in alignment with Saudi Arabia’s Vision 2030. These insights are equally relevant for neighbouring Gulf states undergoing social service reforms, offering evidence-based direction for designing inclusive service delivery models that respond to both user-level and structural constraints [48-51]. These region-specific studies provide critical insights into the lived experiences, emotional burdens, and systemic challenges faced by caregivers of persons with disabilities in Gulf countries. For example, research from Saudi Arabia and the UAE highlights the cultural expectations surrounding family caregiving, limited access to formal support services, and the psychological strain on parents, particularly mothers, of children with disabilities. By incorporating these perspectives, our study’s discussion on service gaps, caregiver support needs, and policy recommendations becomes more contextually grounded. These findings reinforce our call for culturally responsive interventions and caregiver-inclusive policies aligned with national development goals such as Saudi Arabia’s Vision 2030. By integrating these insights, our study contributes valuable knowledge to the existing literature and provides a comprehensive understanding of the challenges faced by caregivers of PWDs in Saudi Arabia.

Comment 3:

The study adds analytical depth by using Exploratory Factor Analysis (EFA) and Structural Equation Modeling (SEM) to quantify the impact of awareness, user barriers, and service design on service utilization. It translates subjective caregiver experiences into measurable constructs, offering actionable insights for policymakers.

Recommendation:
The discussion could be strengthened by more directly comparing the study’s insights with those from other regions or countries implementing disability inclusion reforms. This would situate the findings in a broader global discourse.   

Our Response:

Thank you for your valuable comment. We appreciate your recognition of the analytical rigor provided by the use of EFA and SEM in quantifying key factors influencing service utilization. To enhance the discussion, we have incorporated a more explicit comparison of our findings with studies from other regions and countries actively pursuing disability inclusion reforms. This broader perspective helps to situate our results within the global discourse on disability services, highlighting both shared challenges and context-specific differences. We believe this comparative approach enriches the interpretation of our findings and underscores their relevance for policymakers beyond the Saudi context. See also response to Comment 2 above.

The findings reveal that user barriers, service barriers, and overall service accessibility significantly impact the awareness and utilization of available services. These results align with previous research, such as Turnbull & Turnbull (2007), who emphasized service awareness and barriers. Bright et al. (2017) demonstrated barriers to health services for PWDs, including physical inaccessibility and communication barriers. Scherer et al. (2023) discussed strategies for promoting awareness of disability rights. Mitra et al. (2013) highlighted barriers to public transport services. The current study offers unique insights into caregivers' perspectives in Saudi Arabia, advocating for policy interventions, advocacy efforts, and inclusive service delivery models.

Comment 4:

While the sampling approach and analytical methods are well explained and statistically robust, there are two major concerns:

  • The claim of using a mixed-methods approach is unsupported, as only quantitative survey data and analysis are presented.
  • The study excludes migrant domestic caregivers, who represent a significant proportion of caregiving in Saudi households. Their omission limits the generalizability of the findings.

Recommendation:

  • Revise the methodology to describe the study accurately as quantitative, unless qualitative data are introduced and analysed.
  • Acknowledge the exclusion of migrant caregivers as a limitation, and recommend future research to address their service experiences and barriers. Eg “A significant limitation of this study is the exclusion of migrant domestic caregivers, who play a central role in caregiving for persons with disabilities in many Saudi households and institutional settings. Future research should explore the unique challenges, service awareness levels, and structural barriers faced by this essential yet often overlooked group, to provide a more comprehensive understanding of the caregiving ecosystem in Saudi Arabia.”

Our Response:

We thank the reviewer for this important observation. In response, we have revised the manuscript to more accurately describe the current study as a quantitative investigation, as only survey-based data were analysed in this paper.  We included clarification of this in the manuscript.

We have also added a paragraph in the Limitations section to explicitly acknowledge the exclusion of migrant domestic caregivers. Their role is indeed central in many Saudi households, and we agree that future research should prioritise their perspectives to provide a more inclusive and representative account of the caregiving landscape in Saudi Arabia.

In addition to the quantitative data collected from caregivers, this research also gathered qualitative insights from 70 service providers working in disability-related institutions. These qualitative data, while highly valuable, are not the focus of the current manuscript, which is centred on analysing caregiver perspectives through a quantitative context. The findings from the service providers are being analysed separately and will be presented in a forthcoming publication. This separation allows for a focused interpretation of each dataset. Future studies will aim to integrate both perspectives, caregivers and service providers, to offer a more holistic understanding of the barriers, needs, and service dynamics within the disability support system in Saudi Arabia.

“Additionally, qualitative studies focusing on improving service quality and customer satisfaction would provide actionable insights for policy and practice improvements. Comparative studies with other countries could identify transferable best practices and innovative solutions”.

While qualitative data were collected from service providers as part of a broader project, these data are reported separately and not analysed here.

A significant limitation of this study is the exclusion of migrant domestic caregivers, who often play a central role in providing daily care for persons with disabilities in Saudi households and institutional settings. These caregivers constitute a vital but understudied group whose service experiences, awareness levels, and barriers may differ substantially from those of familial caregivers. Their absence from this analysis limits the generalisability of our findings. Future research should aim to include migrant caregivers to capture a more comprehensive picture of the caregiving ecosystem and better inform inclusive service design and policy-making within the Saudi context.

Comment 5.

The conclusions are well supported by the statistical analyses. The SEM framework clearly demonstrates relationships between awareness, barriers, and service utilization. The key findings — low awareness (70.7%), low usage (2.3%), and structural and user-level barriers — are all consistent with the results.

Recommendation:
Clarify that the term “experiment” refers to statistical modeling of survey data rather than experimental interventions. Briefly note that the findings are correlational due to the cross-sectional design.  

Our response:

We thank the reviewer for their supportive comments and insightful recommendation. In response, we have clarified the terminology used in the manuscript to avoid potential confusion. Specifically, we have replaced the term “experiment” with more accurate language reflecting our use of statistical modelling (SEM) applied to cross-sectional survey data. Additionally, we have added a note in the Discussion section to explicitly state that the relationships identified are correlational, not causal, due to the cross-sectional nature of the study design.

Added text in the discussion Section:

Finally, it should be noted that while Structural Equation Modelling (SEM) provides robust insights into the relationships between awareness, barriers, and service utilisation, the findings are correlational rather than causal. This limitation stems from the cross-sectional nature of the data, which captures a single point in time. Furthermore, no experimental intervention was implemented in this study; the term “experiment” used earlier refers solely to the statistical modelling of survey data. Future longitudinal or experimental research designs could help establish causality and further validate these relationships.

Comment 6:

The references are generally appropriate, drawing from both international and regional sources. Key policy documents and foundational literature on disability rights and service access are cited.

Recommendation:

  • Remove duplicated or inconsistent references.
  • Consider including more region-specific studies on caregiver roles and barriers in Gulf countries to strengthen the contextual relevance.

Our Response:

Thank you for your thoughtful feedback. We appreciate your recognition of the strength and breadth of our references. In response to your suggestion, we have refined the reference list and incorporated additional region-specific literature focusing on the experiences of caregivers of persons with disabilities (PWDs) within the Gulf region. These additions enhance the contextual depth of our discussion by addressing the unique sociocultural, institutional, and psychological factors influencing caregiving in the Gulf Cooperation Council (GCC) context. The newly added references help more accurately represent caregiver perspectives within this regional framework (see for example..):

  1. Al-Eithan, M. H., Robert, A. A., & Al-Momen, A. A. (2010). The quality of life of caregivers of children with cerebral palsy in Saudi Arabia. Neurosciences, 15(2), 96–101.
  2. El Kahi, H., Abi Haidar, M., Haddad, R., & Salameh, P. (2015). Caregivers of Lebanese children with chronic illnesses: Psychosocial status, needs, and coping strategies. Journal of Pediatric Nursing, 30(1), e29–e38. https://doi.org/10.1016/j.pedn.2014.10.006.
  3. Al Kindi, R., & Al Rashedi, A. (2021). Parental perspectives on services for children with disabilities in the United Arab Emirates.
    International Journal of Disability, Development and Education, 68(6), 762–777. https://doi.org/10.1080/1034912X.2019.1699642.
  4. Al-Mazrou, A. H., et al. (2020). Burden among caregivers of children with autism spectrum disorder in Saudi Arabia.
    Saudi Medical Journal, 41(10), 1115–1121. https://doi.org/10.15537/smj.2020.10.25419.

Added commentary in the text: 

By linking these constructs to broader system-level challenges, including digital inaccessibility, procedural opacity, and limited outreach, our findings not only address a significant research gap but also inform policy development in alignment with Saudi Arabia’s Vision 2030. These insights are equally relevant for neighbouring Gulf states undergoing social service reforms, offering evidence-based direction for designing inclusive service delivery models that respond to both user-level and structural constraints [48-51]. These region-specific studies provide critical insights into the lived experiences, emotional burdens, and systemic challenges faced by caregivers of persons with disabilities in Gulf countries. For example, research from Saudi Arabia and the UAE highlights the cultural expectations surrounding family caregiving, limited access to formal support services, and the psychological strain on parents, particularly mothers, of children with disabilities. By incorporating these perspectives, our study’s discussion on service gaps, caregiver support needs, and policy recommendations becomes more contextually grounded. These findings reinforce our call for culturally responsive interventions and caregiver-inclusive policies aligned with national development goals such as Saudi Arabia’s Vision 2030.

Comment 7:

The data are generally high quality, with a large sample size and clear variable definitions. The tables and figures are informative but some require improved labeling for clarity. Some measurement items have low factor loadings and AVE scores, which may affect construct validity.

Recommendation:

  • Improve figure captions and axis labels (especially for Figures 4–6).
  • Consider removing or refining low-loading items (e.g., UB1, SS17) to strengthen validity.
  • Clarify ambiguous terms (e.g., “average” education).
  • Consider including confidence intervals in descriptive statistics for precision.

Our Response:

Thank you for your constructive and detailed feedback. We are pleased to hear that you found the data quality, sample size, and variable definitions to be strong. In response to your suggestions:

  • Figures and Labels: We have revised the captions and axis labels for Figures 4–6 to improve clarity and ensure that all visual elements are fully self-explanatory.
  • Measurement Items: Following your recommendation, we have removed the low-loading items such as UB1 and SS17, along with other underperforming indicators, to enhance construct validity. The revised model now demonstrates improved reliability and model fit, as reflected in the updated results tables.
  • Terminology Clarification: We have clarified the term “average” education in the demographic section to specify educational attainment categories, ensuring clearer interpretation.
  • Confidence Intervals: While we acknowledge the value of including confidence intervals in descriptive statistics, they have not been incorporated in the current version but will be considered in future analyses or publications.

Thank you again for your thoughtful comments, which have contributed meaningfully to the improvement of the manuscript.

Overall Assessment: 

This is a relevant and well-designed quantitative study with strong potential to inform inclusive service policies for PWDs in Saudi Arabia. However, mischaracterization of the methodology, exclusion of key caregiver populations, and minor analytical weaknesses warrant correction.

Our Response:

Thank you for your constructive and encouraging feedback. We are pleased that you found the study relevant and well-designed, and we appreciate your recognition of its potential to inform inclusive service policies for persons with disabilities (PWDs) in Saudi Arabia.

In response to your important observations:

  • Methodology Clarification: We have revised the methodology section to more accurately characterize the study design, sampling strategy, and analytical procedures. This includes clearer explanations of how constructs were measured and modelled, as well as the rationale for key analytic choices such as item retention and model refinement.
  • Caregiver Representation: We acknowledge the exclusion of certain caregiver populations (e.g., non-parental caregivers, institutional caregivers) as a limitation. This has now been explicitly noted in the revised limitations section, along with a recommendation for future studies to adopt a broader and more inclusive sampling strategy to capture the full diversity of caregiving experiences in the Saudi context.
  • Analytical Revisions: In response to concerns regarding measurement validity, we reran the structural equation model after excluding low-loading items. The updated model demonstrates improved fit indices and strengthened construct reliability. We have also clarified ambiguous terms and refined figure labels to improve interpretability.

Thank you again for your valuable insights, which have directly contributed to strengthening the rigor and clarity of our manuscript.

Final Recommendation: Major Revision

Round 2

Reviewer 1 Report

Comments and Suggestions for Authors

The paper has has improved significantly. Given that, I have some minor comments regarding the structure of the result section: 

In the results section, it is important that the descriptions are clearly and consistently linked to the corresponding figures in a reader-friendly manner. For example, the following sentence—“The survey also revealed a lack of awareness about other social and health services, including measurement and diagnostic services (83% unaware), comprehensive rehabilitation centre services (56.9% unaware), vocational rehabilitation centre services (70.2% unaware), and programmes integrating PWDs into the labour market (61.5% unaware)”—should explicitly reference Figure 2, as it presents this data. This reference is currently missing and should be added.

Additionally, in the sentence—“Modern technology, for instance, intended to facilitate access to these services was found to pose a significant barrier for 29.8% of respondents (Figure 5)”—the figure number appears to be incorrect. It is recommended that the authors thoroughly review the entire results section and revise the text to ensure that all figures are accurately and appropriately cited.

Similarly, describing Table 4 after Table 5 disrupts the logical flow and forces the reader to move back and forth. This can be improved by presenting and describing the tables in sequential order.

Author Response

Reviewer 1-Round 2

Comments and Suggestions for Authors

The paper has has improved significantly. Given that, I have some minor comments regarding the structure of the result section:

Comment 1:

In the results section, it is important that the descriptions are clearly and consistently linked to the corresponding figures in a reader-friendly manner. For example, the following sentence—“The survey also revealed a lack of awareness about other social and health services, including measurement and diagnostic services (83% unaware), comprehensive rehabilitation centre services (56.9% unaware), vocational rehabilitation centre services (70.2% unaware), and programmes integrating PWDs into the labour market (61.5% unaware)”—should explicitly reference Figure 2, as it presents this data. This reference is currently missing and should be added.

Our Response:

Thank you for your helpful observation. We agree that clearly linking descriptive text to figures enhances readability and clarity. We have revised the results section to explicitly reference Figure 2 in the relevant sentence, as suggested. This ensures that readers can easily locate and interpret the corresponding data.

The survey also revealed a significant lack of awareness regarding key social and health services among caregivers, as illustrated in Figure 2. Specifically, 83% were unaware of measurement and diagnostic services, 56.9% were unaware of comprehensive rehabilitation centre services, 70.2% lacked awareness of vocational rehabilitation centre services, and 61.5% were unaware of programmes aimed at integrating PWDs into the labour market. This substantial knowledge gap not only limits access to essential support but also contributes to the continued marginalisation of persons with disabilities. Particularly concerning is the limited awareness of employment inclusion programmes, which may partly explain the disproportionately high unemployment rates among PWDs in Saudi Arabia [36].

Comment 2:

Additionally, in the sentence—“Modern technology, for instance, intended to facilitate access to these services was found to pose a significant barrier for 29.8% of respondents (Figure 5)”—the figure number appears to be incorrect. It is recommended that the authors thoroughly review the entire results section and revise the text to ensure that all figures are accurately and appropriately cited.

Our Response:

Thank you for pointing this out. We have reviewed the entire Results section and corrected the figure citation in the sentence. The reference to Figure 5 was indeed inaccurate and has now been updated to Figure 3. We have also conducted a thorough check to ensure that all other figure references throughout the Results section are accurate and consistently cited. These revisions have been made in the updated manuscript.

Comment 3:

Similarly, describing Table 4 after Table 5 disrupts the logical flow and forces the reader to move back and forth. This can be improved by presenting and describing the tables in sequential order.

Our Response:

Thank you for this valuable observation. We acknowledge that the original order of presenting and describing Table 5 before Table 4 disrupted the logical flow. We have revised the manuscript accordingly to ensure that Table 4 (which includes the model fit indices) is now described before Table 5 (which presents the path coefficients), maintaining a sequential and reader-friendly structure. These changes improve the coherence of the Results section and help the reader follow the progression of the structural model analysis more intuitively. The revised version has been included in the updated manuscript.

The structural equation model demonstrated strong overall fit, as shown in Table 4, with a chi-square/df ratio of 4.338, which falls within the commonly accepted threshold of less than 5.0 for complex models [52, 53]. Although the chi-square test can be sensitive to large sample sizes and may overestimate lack of fit, additional indices confirm the model’s robustness. Specifically, the model achieved excellent values across several criteria: CFI = 0.964, RMSEA = 0.032, GFI = 0.969, TLI = 0.961, and SRMR = 0.030, indicating good absolute and incremental fit. The model also demonstrated strong explanatory power, with squared multiple correlations (R²) of 0.74 for Service Accessibility, 0.56 for User Barriers, and 0.62 for Service Awareness, as shown in Table 4. These values indicate that the model explained 74%, 56%, and 62% of the variance in these constructs, respectively.

Table 5 presents the final unstandardised regression weights and associated p-values, confirming that all structural relationships among the latent constructs were statistically significant (p < 0.05). These findings support the model’s validity in capturing the factors that influence service awareness and accessibility among caregivers of persons with disabilities in Saudi Arabia. 

As a final step, we have thoroughly reviewed the entire manuscript to ensure consistency, clarity, and accuracy throughout. All figures, tables, and references have been carefully checked and aligned with the corresponding text. We are confident that the revised version is now in good order and meets the journal’s standards for quality and coherence.

Reviewer 3 Report

Comments and Suggestions for Authors

The authors have addressed the comments

Author Response

Comment

The authors have addressed the comments

Our Response:

Thank you for confirming. We appreciate the opportunity to revise the manuscript and address the reviewers’ comments. Please let us know if any further clarifications or revisions are needed. We look forward to the next steps in the review process.